# Assessing costs of Indonesian fires and the benefits of restoring peatland

L. Kiely[1,6 ✉], D. V. Spracklen [1], S. R. Arnold[1], E. Papargyropoulou[2], L. Conibear[1], C. Wiedinmyer [3], C. Knote [4] & H. A. Adrianto[1,5]

Deforestation and drainage has made Indonesian peatlands susceptible to burning. Large fires occur regularly, destroying agricultural crops and forest, emitting large amounts of $CO_2$ and air pollutants, resulting in adverse health effects. In order to reduce fire, the Indonesian government has committed to restore 2.49 Mha of degraded peatland, with an estimated cost of US$3.2-7 billion. Here we combine fire emissions and land cover data to estimate the 2015 fires, the largest in recent years, resulted in economic losses totalling US$28 billion, whilst the six largest fire events between 2004 and 2015 caused a total of US$93.9 billion in economic losses. We estimate that if restoration had already been completed, the area burned in 2015 would have been reduced by 6%, reducing $CO_2$ emissions by 18%, and $PM_{2.5}$ emissions by 24%, preventing 12,000 premature mortalities. Peatland restoration could have resulted in economic savings of US$8.4 billion for 2004–2015, making it a cost-effective strategy for reducing the impacts of peatland fires to the environment, climate and human health.

[1] School of Earth and Environment, University of Leeds, Leeds, UK. [2] Sustainability Research Institute, School of Earth and Environment, University of Leeds, Leeds, UK. [3] CIRES, University of Colorado, Boulder, CO, USA. [4] Ludwig-Maximilians University, Munich, Germany. [5] IPB University, Bogor, Indonesia. [6] Present address: Department of Chemical and Environmental Engineering, University of California, Riverside, CA, USA. ✉email: lkiely@ucr.edu

Southeast Asia contains 25 million hectares of tropical peatland, mostly in Indonesia[1]. Indonesian peatlands store an estimated 57 Gt of carbon, 55% of the world's tropical peatland carbon[1,2], contain substantial biodiversity value[3] and support local livelihoods through provision of ecosystem services[4]. Deforestation and drainage associated with the expansion of small-holder and industrial-scale agriculture[5–8] has caused extensive degradation of Indonesian peatlands[9], increasing the risk and vulnerability to fire.

Indonesian peatlands rarely experienced fire until recent decades[10]. Large fires are now a regular occurrence[11], with the two largest fire events on record occurring in 1997 and 2015[12,13]. Fires generally occur during periods of drought[14], and are closely linked with land-use change[15,16]. Drainage and deforestation of extensive areas of peatland in Indonesia make the naturally fire-resilient peatland susceptible to fire[17,18]. Fires lit to clear land can burn out of control and spread into degraded forests and peatlands, particularly during El Niño years.

Peatland fires cause large $CO_2$ emissions[19–21], contributing substantially to Indonesia's greenhouse gas emissions[22]. Fires in Equatorial Asia (mostly Indonesia) were responsible for 8% of global fire carbon emissions in 1997–2016[23]. Fires also emit large quantities of fine particulate matter ($PM_{2.5}$) and other pollutants, resulting in poor air quality and negative health effects[24–26]. Fires destroy agricultural land and forest resources, while haze can disrupt transport, tourism, and trade, slowing the economic performance of a region[27]. The losses and damages caused by landscape fires cost billions of US$ and exceed 1% of gross domestic product (GDP) in countries where fire is prevalent[28]. During the El Niño of 1997–1998, fires burnt across 8 million hectares of Indonesia resulting in losses estimated at between US$4.5 billion[29] and US$19.7 billion[30] through damage to agriculture and forest, $CO_2$ emissions and health impacts from exposure to fire haze. In comparison, fires in the Amazon during the same year burnt 5.9 million hectares, causing economic losses of US$9.5 billion through damage to agriculture and forest, $CO_2$ emissions and health impacts from exposure to fire haze[31]. More recently, the 2015 fires in Indonesia are estimated to have cost US$16.1 billion[22] whilst the 2019 fires cost US$5.2 billion[32] in damages and economic losses to agriculture, forestry, trade, tourism, transportation, manufacturing and the environment, and through the costs of fire suppression, short-term health impacts and school closures. These estimates did not include the economic costs of long-term health impacts from exposure to haze from the fires, meaning the actual cost is likely to be much higher[33].

Due to the detrimental impacts of fires, a moratorium on any new land conversion on peatland was brought into effect in Indonesia in 2011[34], and in 2016 the Peatland Restoration Agency (Badan Restorasi Gambut, BRG) was established to restore and re-wet 2.49 million hectares of degraded peatland[35]. Fires are more likely to occur on degraded land than in protected areas of forest[18,36], and drainage canals can make fires 4.5 times as likely[17]. Controlling land-use change and blocking drainage canals on peatland should therefore reduce fire and associated emissions. Since the spread of peatland fires is dependent on the water level[37], re-wetting peatlands can be important for controlling fires. However, restoring degraded peatlands is challenging and large-scale efforts to restore tropical peatlands are in their infancy[38].

Recent studies have found that the moratorium on land conversion may not have been effective in reducing deforestation or fires[39,40]. There have so far been no comprehensive estimates of the potential impacts of peatland restoration initiatives on fire occurrence. Crucially, large-scale restoration efforts to address fire-related problems lack a cost-benefit analysis[41].

To help address this gap, we estimated the impact of peatland restoration on fire and the associated loss and damages caused by fire. First, we estimated the loss and damages caused by Indonesian fires in recent years, finding US$93.9 billion in economic losses from the six largest fire events. Second, we estimated the losses and damages under a scenario where 2.49 million hectares of degraded peatland had been restored, finding a reduction in economic losses of US$8.4 billion. By contrasting this benefit against the estimated costs of restoration, our analysis demonstrates that the benefits of effective peatland restoration will outweigh the cost of restoration, and provides evidence to support ongoing peatland restoration efforts.

## Results and discussion

**Economic losses and damages due to fires**. We estimated the economic costs of Indonesian fires, focusing on the six largest dry season (August–October) fire events from 2004 to 2015 (Fig. 1). Previous estimates of fire cost have included different economic losses[22,30], with health impacts, $CO_2$ emissions and damage to crops, forests, and plantation causing the majority of the total costs[9]. For this study we have therefore focused on these three main contributing sectors (Supplementary Table 1) and we do not attempt to estimate the other costs and impacts of fire.

Economic losses due to damages to agriculture, plantation, natural forest and other land covers were estimated by combining the area burnt with the net present value of each land use. The greatest cost from damages to land cover occurred in 2006 (US$11 billion) and 2015 (US$9.4 billion), with costs in other years between US$4 billion and US$7 billion. The damages to plantation crops and natural forest made up the majority of these losses (Fig. 1). In using one value for each land use, consumer and producer surplus have not been considered, and the true economic losses due to fires may differ.

The costs associated with $CO_2$ emissions were estimated by combining $CO_2$ emissions from a fire emission inventory[24], with the 2009–2020 average value of $CO_2$ from the European Union Emissions Trading System. The 2015 fires resulted in the largest $CO_2$ emissions (962 Tg) with an imputed damage value of US$11.3 billion. Our estimate of the $CO_2$ emissions from the 2015 fires lies within the range from previous studies (547–1100 Tg)[13,19,42]. In other years $CO_2$ emissions varied between 272 Tg and 542 Tg with imputed damage values of US$3.2–6.4 billion.

The economic cost associated with the health effects caused by exposure to haze from fires was calculated based on the number of Disability Adjusted Life Years (DALYs) caused by smoke exposure multiplied by the economic value of a DALY. The 2015 fires caused the largest economic losses associated with health impacts (US$7.3 billion), with US$5.7 billion losses for Indonesia, US$1.3 billion for Malaysia and US$0.3 billion for Singapore. In other years, the total health-related economic losses were US$1.8–3 billion.

The total economic losses caused by fire was greatest for the 2015 fires, with economic losses of US$28 billion (Fig. 1). Of the total economic losses and damages, 33% were due to land-cover damage, 40% from the imputed damage value associated with $CO_2$ emissions, and 26% from the economic losses associated with long-term health costs. In other years total economic losses and damages were US$9.1–20.4 billion, with the damage to land cover contributing around half of the total, and $CO_2$ emissions contributing around a third. In 2015, total losses and damage were equivalent to 3.3% of Indonesian Gross Domestic Product (GDP). In other years, loss and damage were equivalent to 1.1–2.4% of Indonesian GDP, meaning Indonesia's economy is one of the most heavily impacted by fire[28]. In 2015 severe drought caused fires to burn deeper into the peat resulting in

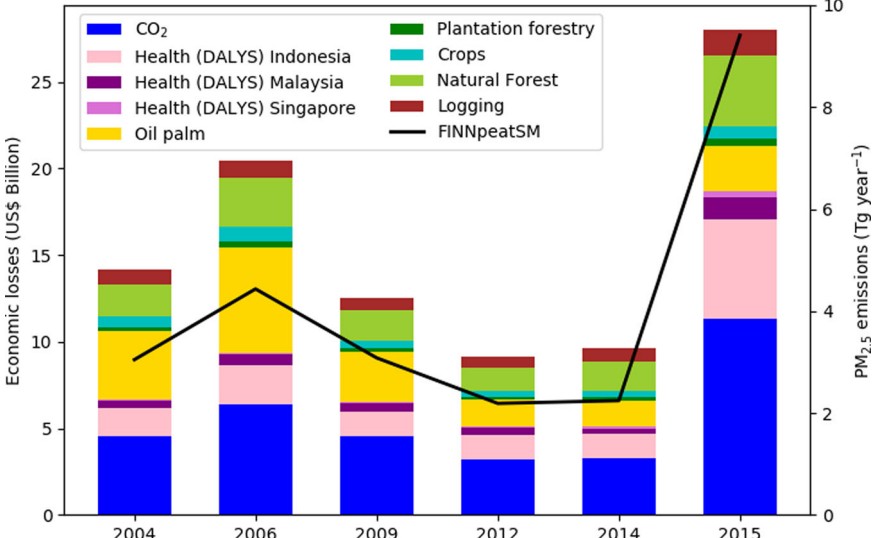

**Fig. 1 The estimated economic losses and damage caused by Indonesian fires.** Economic losses, in US$ billion, split by category, are shown by the bars. The health costs are estimated based on Disability Adjusted Life Years (DALYs) and split by the country being affected. Economic losses connected to land-use damage are split by land-use. Total dry season (August–October) PM$_{2.5}$ emissions from the FINNpeatSM fire inventory are shown for each of the years by the black line.

larger emissions per area burned[24], and the costs associated with $CO_2$ and PM$_{2.5}$ emissions were greater.

The World Bank[22] estimates the economic cost of the 2015 fire event to be US$16.1 billion, less than suggested in our study, largely due to the lack of long-term health impacts in the World Bank estimate. The estimate of costs due to damage and loss of agriculture and forest in the World Bank study (US$8.7 billion) is similar but smaller than our estimate (US$9.4 billion), despite them also including equipment damage. This could be because the World Bank used burned area from the Global Fire Emissions Dataset, which has previously been found to be underestimated in the region[43]. The cost from sectors not included in our study has been estimated at US$3.4 billion[22].

**Reduced costs of using fire for land clearing**. Landowners use fire to clear land because it is can be easier and cheaper than other methods such as mechanical clearance[44]. Guyon and Simorangkir[45] find that clearing forest without the use of fire has increased labour and equipment cost. We estimate the reduced costs of using fire to clear forest, compared to other mechanical clearance options, to be up to US$1.2 billion across the 6 years studied. This includes both forest that has been intentionally cleared as well as forest destroyed by fires that escape into surrounding land, so will be an overestimate of the reduction in costs. Despite this, the economic losses caused by damage to agriculture, which totals US$23.5 billion over the six years we study, are much greater than the reduced costs of using fire versus other land clearance options. Despite the economic losses caused by fire (e.g. due to damages to agricultural land) exceeding the economic savings of using fire to clear land, small-scale farmers may not have access to mechanical equipment[44]. This means many farmers may have little option but to continue to use fire. Morello et al.[46] suggested that for the Amazon, a policy of subsidising mechanical clearing equipment improves the effectiveness of banning fire. Mechanical clearing is an effective way of maintaining existing agricultural land and could be more widely adopted if equipment was more widely available[44].

**Fires in protected areas**. Peatland restoration involves blocking drainage canals to restore water levels and re-establishing vegetation cover[47]. Large-scale peatland restoration in Indonesia has just begun, and it is too early to measure the effect on fire[48]. Instead, we used fires observed within protected areas as a proxy for fire occurrence on restored peatland. Peatland in protected areas is largely undrained and still covered in vegetation but is still subject to drought and anthropogenic pressures meaning that protected areas experience degradation, deforestation[49], and fire, albeit at a lower rate than surrounding unprotected land[49–52]. Protected areas therefore provide a useful indication of the susceptibility of restored and re-wetted peatlands to fire under existing climate and anthropogenic pressures.

We compared the occurrence of fire inside protected areas in Indonesia with the surrounding area. For each year we calculated the ratio of peatland burned area inside and outside of protected areas. Comparing directly with the surrounding area avoids issues connected to bias in the location of protected areas[50]. We find that protected areas typically reduce the occurrence of fire, though the effects are variable depending on location and protected area type (Fig. 2, Supplementary Table 2) as found previously for both deforestation[49–51,53] and fire[52]. National Parks in Kalimantan result in the greatest reduction in fire. This is likely due to the fact that National Parks are generally larger than other types of protected areas, reducing outside influences such as drainage, although the more effective management of National Parks could also be important (see supplement). We find that for National Parks in Kalimantan total dry season burned area on peatland in 2004–2015 was reduced by 37–79% compared to the surrounding areas, depending on the year. Protected areas are less effective at reducing fire in drought years (e.g. 2015) compared to non-drought years.

Depth of peat burn and emissions from peat fires depends on water levels in peatlands, which can be heavily impacted by land-use change and drainage. To explore how protection of peatlands can modify water storage, we compared soil moisture inside and outside of protected areas. We used soil moisture from the Soil Moisture Active Passive product (SMAP) which is available for 2015 onwards. Monthly average soil moisture in August–October 2015 was 48–57% greater inside National Parks compared to outside, likely due to reduced drainage inside the protected areas. Greater soil moisture will reduce the burn depth of fires, resulting in lower emissions from fire within protected areas.

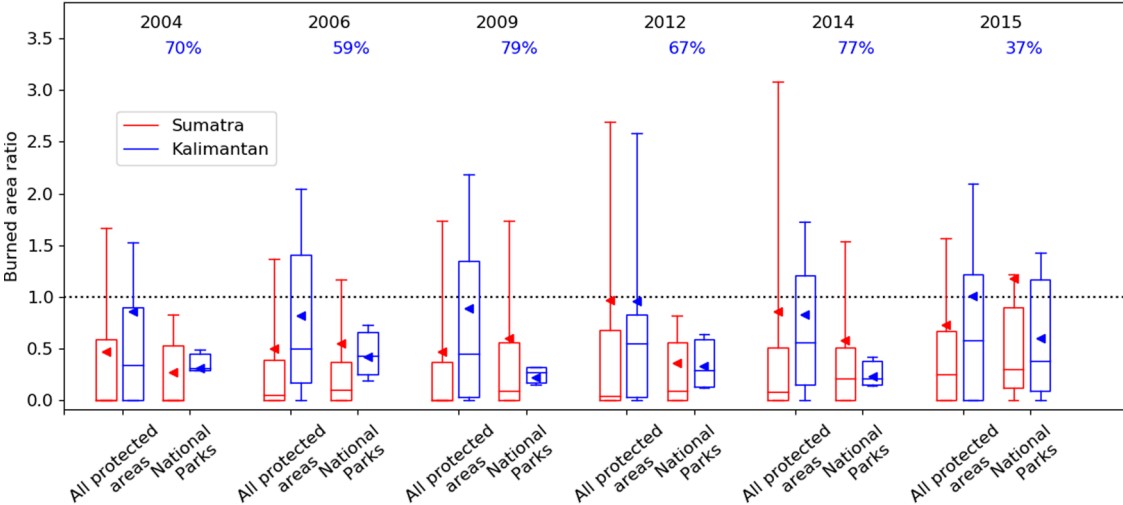

**Fig. 2 The ratio of peatland burned area inside protected areas to outside of protected areas.** For each protected area we compare fraction of peatland burned inside to outside (within 0.25° latitude and longitude of the protected area). Results are shown for Sumatra and Kalimantan for all protected area categories ($n = 31$ and 22, respectively) and for National Parks ($n = 12$ and 6, respectively) in each year. The box shows the upper and lower quartiles, the whiskers show the 95th percentiles, the lines show the median and the triangles the mean. The average percentage reduction in burned area inside the National Parks in Kalimantan in each year is shown in blue text.

**Effects of peatland restoration on emissions.** We used the reduction in fire occurrence and increase in soil moisture in protected areas in Kalimantan to estimate the potential reductions in fire emissions that would have been achieved under a policy of peatland restoration. We estimated the burned area and emissions under a scenario where 2.49 Mha of degraded peatland was restored, the area planned for restoration by the Indonesian peatland restoration agency. Fires are more likely to occur in areas made susceptible to fire by land-use change and drainage[54,55], and so the locations of past fires are likely to be where future fires occur; areas of degraded peatland in Kalimantan have been found to have burned up to eight times between 1990 and 2011[56]. We therefore selected locations for peatland restoration by identifying peatlands with the greatest $PM_{2.5}$ emissions during 2004–2015. We assumed restoration areas of ~500 km$^2$ (50,000 ha) based on the evaluation of protected areas (see methods and Supplementary Table 4), with 2.49 Mha equivalent to 51 restoration areas. Regions selected for restoration are all located in Kalimantan and Sumatra, with the majority in southern Central Kalimantan or South Sumatra (Fig. 3). For each of the years we studied between 2004 and 2015, we calculated the change in fire and associated emissions that would have occurred if 2.49 MHa had been restored prior to the occurrence of fires. For each year, we recalculated fire emissions with the burned area and soil moisture inside restored peatland areas scaled by the ratios of burned area and soil moisture inside and outside of National Parks in Kalimantan for that year.

To assess the uncertainty around our treatment of fire on restored areas, we explored the effect of two other scenarios on fire emissions. For one option, we assumed all fires are prevented on the restored peatland. Studies have found that fires continue to occur after peatland restoration[57,58], suggesting that 'no fire' is unlikely to be achieved by restoration. However, it provides a reference for a natural state in the absence of any anthropogenic pressures and is the maximum reduction that could theoretically be achieved under restoration. Our final scenario assumed all peat fires are prevented, with only surface vegetation fires and associated emissions remain. This 'no peat fire' scenario could occur if peatlands are re-wetted, and remain saturated throughout the dry season preventing the peat from burning, but fires continue on the surface.

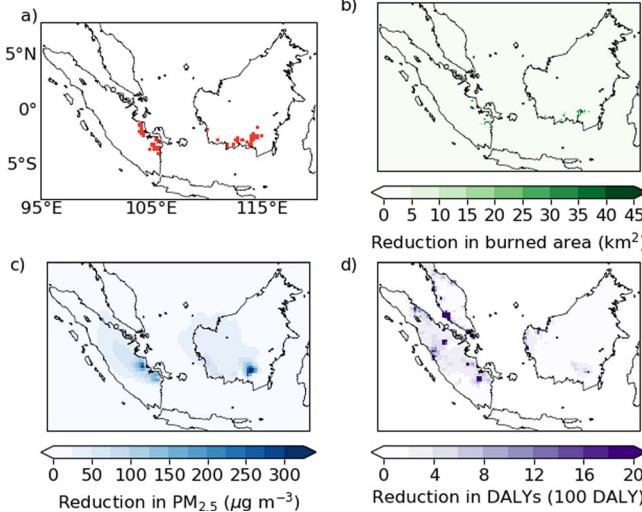

**Fig. 3 The potential impacts of peatland restoration on August–October 2015 fires.** Locations of peatland restored in this study are shown in red (**a**). Reductions in burned area (**b**), average $PM_{2.5}$ concentrations (**c**), and Disability Affected Life Years (DALYs) from $PM_{2.5}$ exposure (**d**) due to peatland restoration are shown by the green, blue, and purple colour scales, respectively.

In 2015, 15% of the total burned area in Sumatra and Kalimantan occurred on areas selected for restoration. Our analysis suggests restoration to the level of National parks would have reduced peatland area burned in 2015 by 37% (Fig. 2), resulting in an overall reduction in area burned across Kalimantan and Sumatra by 6%. Restoration reduces August–October $PM_{2.5}$ emissions by 24% from 9.45 to 7.27 Tg, and $CO_2$ emissions were reduced by 18% from 962 to 790 Tg (Fig. 3). The percentage reduction of $CO_2$ is less than of $PM_{2.5}$ as the latter has a greater contribution from peat fires, which are reduced by both reduction in burned area and burn depth. Restoration causes smaller reductions in other years, between 8–15% for $PM_{2.5}$ and 6–11% for $CO_2$ (Fig. 4). In comparison, the moratorium on new agricultural concessions on peatlands has been estimated to have reduced $CO_2$ emissions by 2.5–7.2%[59].

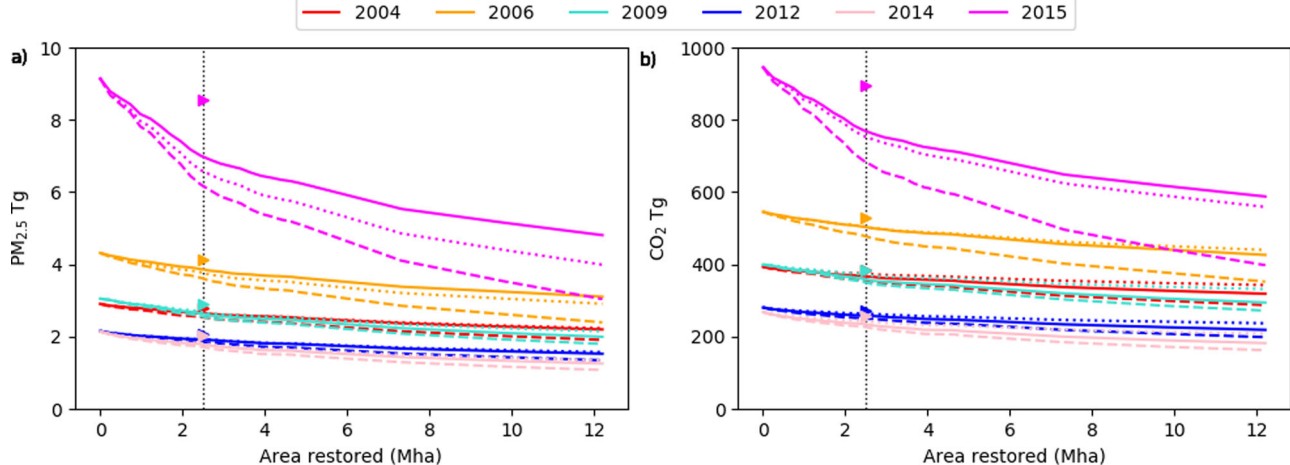

**Fig. 4 Reduction in fire emissions under peatland restoration scenarios.** Emissions of PM$_{2.5}$ (**a**) and CO$_2$ (**b**) under different peatland restoration scenarios (National Parks: solid line, No peat fires: dotted line and No fire: dashed line). The triangles show the emissions when 2.49 Mha is randomly located, under the National Parks scenario. The black dotted vertical line shows 2.49 Mha restored.

Restoring peatland would also reduce the carbon emitted through peatland oxidation that is associated with degradation[60–62]; however, this benefit has not been included in our study.

Figure 4 shows PM$_{2.5}$ and CO$_2$ emissions under our different peatland restoration scenarios. The 'no fire' scenario on restored peatlands results in the largest reduction in emissions (Fig. 4). Under the scenario of 2.49 MHa restored, the 'no fire' scenario reduced PM$_{2.5}$ emissions in 2015 by 32% and CO$_2$ emissions by 26%. In other year emissions were reduced by 9–19%. The 'no peat fire' scenario gives a similar reduction in CO$_2$ emission to the National Parks scenario, but for PM$_{2.5}$ emissions it varies. In 2015, when intense drought meant that peat fires burnt deep into the ground, the 'no peat fire' scenario is more effective than the National Parks scenario, with a 27% reduction in PM$_{2.5}$ emissions compared to a 23% reduction under the National Parks scenario. For other years when peat fires had a smaller contribution to emissions, the 'no peat fire' and National Parks scenarios for restoration show similar reductions in emissions: 5–14% and 6–15%, respectively (Fig. 4). This suggests that for strong drought years re-wetting peatland to prevent fires from burning into the peat is the most effective action. For less intense drought years, reducing the number of fires and area burnt could be more important. The National Parks restoration scenario results in the smallest emissions reduction in most years (Fig. 4), and we use this scenario in our estimates of the benefits of peatland restoration. There is currently little data on the susceptibility of restored peatland to fire, and the effectiveness of peatland restoration interventions are limited to small-scale trials[48]. If peatland restoration is less effective, emission reductions would be lower than we estimate here.

We used a regional atmospheric chemistry model to simulate the impacts of peatland restoration on regional air quality in 2015. Reduced emissions under peatland restoration result in average PM$_{2.5}$ concentrations across the domain being reduced by 28% (from 76 to 55 µg m$^{-3}$) and population-weighted PM$_{2.5}$ being reduced by 26% (from 27 to 20 µg m$^{-3}$). We estimated the number of excess deaths in the region resulting from exposure to PM$_{2.5}$ from fires is reduced by 11,914 (21%), from 55,819 to 43,905 with peatland restoration. The number of Disability Affected Life Years (DALYs) caused by exposure to PM$_{2.5}$ is reduced by 0.46 million, from 2.19 to 1.72 million. While the reduction in PM$_{2.5}$ concentration is greatest near the locations of the restored land, the reduction in exposure and associated DALYs is more regionally dispersed (Fig. 3). For other years we

estimate restoration reduces the number of DALYS by 17,000–94,000.

**Potential for scaling up peatland restoration.** Indonesia has around 21 Mha of peatland, with 13 Mha in Sumatra and Kalimantan[1,63] and 11.5 Mha of this is estimated to have been degraded[5]. We explored how the benefits of peatland restoration would likely change with the scale of restoration (Fig. 4). Under each scenario, the peatland with the greatest PM$_{2.5}$ emissions over 2005–2015 period are prioritised for restoration first. PM$_{2.5}$ and CO$_2$ emissions decrease steeply as the area of peatland restoration is expanded. Although the 2.49 Mha of peatland the government plans to restore results in a substantial emission reduction, further reductions would still occur if more land is restored, particularly in a high fire year such as 2015. Restoration of all peatlands results in a 54% reduction in PM$_{2.5}$ emissions when peatlands are restored to the state of National Parks and a 77% reduction under the 'no fire' scenario. Peatland restoration faces ecological, political, economic, legal, social, and logistical challenges[47], creating substantial barriers that may limit the potential for larger-scale peatland restoration in Indonesia.

Fires are heavily concentrated in regions of peatland degradation and land-use change. In 2015, 53% of fire detections occurred on peatlands which covered only 12% of the land, with the greatest fire detection over degraded peatlands[55]. Prioritising areas for peatland restoration is therefore important. We selected locations with the greatest emissions from fires in previous years, which optimised the reduction in emissions. Randomly allocating the 2.49 MHa of restoration reduces emission reductions by more than half (Fig. 4) demonstrating that targeting restoration is important if benefits are to be maximised. Carbon emissions are greatest the first time a peatland burns and typically decline with subsequent fires[56]. Restoring unburned peatlands in areas of high fire risk will therefore lead to the greatest reduction in emissions. At the scale of our analysis, areas designated for restoration will include both burned and unburned peatland.

**Economic costs and benefits of peatland restoration.** Peatland restoration reduces fire occurrence, leading to economic benefits in the form of reduction in losses and damages due to fire. Figure 5 shows the reduction in the losses and damages due to fires if 2.49 Mha of peatland had been restored prior to the 2004–2015 fires. The total reduction over the 6 years studied is estimated to

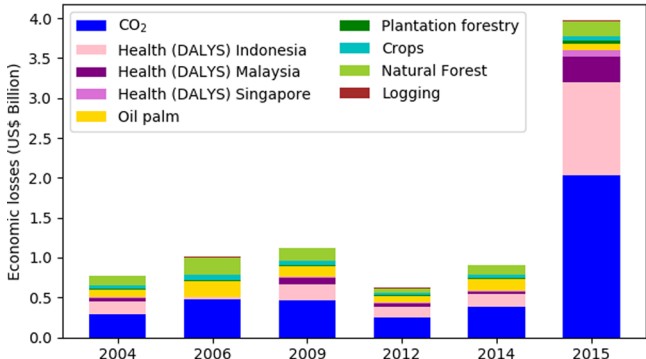

**Fig. 5 The estimated reduction in economic losses and damages caused by fires after peatland restoration.** Bars show the reduction in economic losses, in US$ billion, split by category. The Disability Affected Life Years (DALYs) costs are split by the country being affected.

be US$8.4 billion, with the largest reduction in losses and damages in 2015 (US$4.0 billion). We estimate that peatland restoration would have reduced fire losses and damages by 9% across all years, with a 14% reduction in 2015. The reduction in $CO_2$ emissions contributed the largest reduction in cost overall the years (46%) followed by the reduction in health-related losses (30%) and the reduction in land-cover losses (24%). Other haze related costs such as disruptions to transport and tourism are also likely to be reduced with the reduction to $PM_{2.5}$ emissions, but have not been considered in this study.

We have treated each year individually, and calculated the reduction in costs that could have been achieved if the peatland had been restored prior to each fire event. This provides an indication of the potential savings that restoration could provide for similar fire events in future years. Predicting the cost of fires in the future under business as usual and peatland restoration scenarios is challenging, due to the complex combination of meteorological and anthropogenic drivers of fire. In addition, a range of physical, social and economic feedbacks in the system[21] further complicate the response and have not been assessed here.

The World Bank[22] suggest that restoring 2 Mha of peatland would cost US$1.9 billion, considerably lower than our estimate of economic benefit. If we account for the buy-back value of agricultural land and plantations within the restored areas, costs of restoration rise. Of the land restored in this study, 0.3 Mha is on oil palm concessions, 0.4 Mha is on wood fibre concessions and 0.6 Mha is other agricultural crops. Depending on the land value we estimate a onetime buy back of the land suggested for restoration in this study would cost US$1.3–4.1 billion, bringing the total cost of restoring 2.49 Mha of peatland to US$3.2–6 billion. Another study estimates that restoration costs would exceed US $4.6 billion[64]. Using either value, the cost associated with restoring peatland is less than the associated reduction in fire-related costs.

The cost of peatland restoration may not have to be fully borne by Indonesia. A reduction in Indonesian fires yields health benefits across Indonesia, Malaysia and Singapore. Lin et al.[65] suggests that Singaporeans are willing to pay US$643.5 million for the health benefits of reduced fire. Carbon emissions and climate change are also a global problem. Our analysis further confirms the need for fire to be considered in Reduced Emissions from Deforestation and Degradation (REDD+) programs[66].

There are a range of uncertainties in our analysis. Estimates of the losses and damage to crops, plantation and forest conflate uncertainties in burned area, uncertainties in land-use mapping and the value of different land covers. Estimates of economic costs of $CO_2$ combine uncertainties in burned area, biomass consumption

linked to vegetation loads and burn depth and emission factors[67], as well as the damage value associated with $CO_2$. Emission estimates do not account for declining emissions from recurrent fires on drained peatlands[56]. Health impact estimates combine uncertainty in exposure to particulate matter with uncertainty in concentration response functions and economic costs of health impacts. Despite this large range of uncertainties our estimates are consistent with previous estimates of emissions[23,24,68], population exposure to particulate matter[25,69,70], associated health impacts[24–26], and economic costs[22].

Our analysis aims to cover the main economic losses and damages due to fire, but we have not been able to estimate important impacts of fires on biodiversity, employment, subsistence livelihoods, or regional climate. For this reason we are likely to underestimate both the economic costs of fires and the benefits of peatland restoration. Forests and peatlands of Sumatra and Borneo are an important biodiversity hotspot[71] and forest degradation caused by fire can reduce the biodiversity value of forests[66]. Loss and degradation of forests also impacts regional climate increasing local temperature, reducing regional rainfall[72] and degrading downstream water availability and quality[73]. Forest fires also damage non-timber forest products[74], which make an important contribution to local livelihoods. Our assessment of the health impacts of fire pollution excludes impacts on mental health and other wider health impacts[75] and does not include the public health costs associated with treatment of health impacts. Conversely, fire is an important land management method employed across Indonesia and zero-burn policies can have a negative impact on livelihoods[76]. Future cost-benefit analysis needs to assess a wider range of impacts to give a more complete picture of the costs of fires and benefits and trade-offs connected to peatland restoration.

**Implications for policy.** Our research has important implications for land management and land-use policy in terms of (i) the loss and damages caused by the use of fire as a land clearing technique, (ii) the cost effectiveness of peatland restoration as a fire prevention strategy, and (iii) the conditions under which peatland restoration can deliver the maximum environmental, economic, and health benefits.

We have quantified the losses and damages associated with fire and show they outweigh the savings made by farmers and land managers using fire instead of more expensive mechanical land clearance. Reduced costs of land clearance is often stated as a reason for clearing land with fire[41,44] but potential escape of fire and damage to crops is rarely considered as an incentive to reduce the use of fire[77,78]. Attribution of fire is difficult due to fire spread, overlapping land claims, and impacts of drainage on neighbouring land[79], complicating efforts to effectively direct fire suppression efforts. Local support of fire reduction schemes is a key factor to their success[47,80], and schemes need to identify incentives and sanctions which are important to local people[78].

Our analysis shows that peatland restoration is a cost-effective strategy for prevention of peatland fire in support of existing policies of the Indonesian government. The economic benefits in the form of reduction of fire-related losses and damages, linked to $CO_2$ emissions, long-term health impacts and damage to land cover, outweigh the cost associated with peatland restoration. The benefits of peatland restoration extend from local reductions in property loss, through regional benefits to air quality and public health to global benefits from reduced $CO_2$ emissions. The benefit of restoration depends on the amount of land restored and where the restoration occurs. Restoration should be targeted to areas which have proven to be most susceptible to fires in the past,

which may not always be the case for the current restoration plan[35].

The different fire scenarios we have considered for restored peatland show the variability in possible restoration benefits. Detailed monitoring of peatland restoration and the effect on fires is needed to inform the design of future interventions and improve the cost-benefit analysis of restoration. Restoration can include canal blocking to re-wet the peat and revegetation of degraded peatlands. Re-wetted peatland is unlikely to revegetate naturally[64] and so revegetation should be included in restoration plans. Revegetation can be expensive and current plans include reforesting only 27% of the restoration area[35]. Re-wetting and vegetation of peatlands are some of the most controversial solutions to peatland fire, with a major disconnect between resource users and policy makers[81]. Effective communication of the substantial health benefits delivered by peatland restoration could help build stronger support for restoration interventions from a wider spectrum of stakeholders[82]. Peatland restoration is likely to face serious socio-economic and cultural challenges that will constrain the scale of restoration that can be achieved. Engaging with stakeholders from the outset to understand and mitigate potential farmer and landowner concerns is a crucial element of peatland restoration[83]. Our work also shows the importance of preventing degradation of intact peatlands. Indonesia has a moratorium on deforesting primary forest, however this only covers 32% of Indonesia's peatlands, leaving many vulnerable[84]. We demonstrate that protected areas, particularly National Parks in Kalimantan, are effective at reducing fire and associated emissions from peatlands. This provides evidence to support establishment and maintenance of protected areas in Indonesia as an effective fire management tool. Our analysis found varied effectiveness of protected areas at reducing fire, highlighting the need for effective management of protected areas[85] and peatland restoration. Although emissions are dominated by fires on peatlands, fires on mineral soils contribute an important fraction of total emissions that also need to be addressed. Forests on mineral soils also contain high levels of biodiversity[86] providing further reason for their protection from deforestation and degradation.

In conclusion, we demonstrate the substantial national and international benefits of restoration (including both re-wetting and revegetation) of degraded peatlands in Indonesia. Our work confirms that benefits of restoration outweigh the costs providing evidence to support Indonesia's plans to restore 2.49 Mha of degraded peatland. We show that a more ambitious programme of restoration would yield even greater benefits, especially if restoration was targeted to areas proven to be susceptible to fires in the past, in order to maximise the fire prevention and environmental, health and economic benefits of peatland restoration. Increased drought frequency and consequently greater fire risk across Indonesia under future climate change[87] creates even stronger urgency for ambitious and effective peatland restoration.

## Methods

**Fire emissions**. We used fire emissions and burned area from FINNpeatSM (Supplementary Fig. 1), an extension of the Fire Inventory from NCAR (FINNv1.5)[88]. FINNpeatSM emissions were created specifically for Indonesian fires using recently calculated emissions factors for Indonesian peat fires and a peat burn depth scaled according to the surface soil moisture. The location and area burned by fires in FINN is based on MODIS hotpots. This emission inventory is described further in Kiely et al.[89] and has been comprehensively evaluated for all the years explored here[24]. Fire emissions inventories often either exclude tropical peat fires[88] or underestimate emissions[23], due to difficulties in determining underground fuel consumed and tropical peat emission factors. Total FINNpeatSM PM$_{2.5}$ emissions in 2015 are a factor of 3.5 times those given by FINNv1.5, and 1.7 times those given by GFED4s[89]. We focus our analysis on the major fire season in Indonesia, and report values for fire occurring from August 1 through to October 31 for each year.

**Health impacts**. We estimate DALYs from fires using the same method as Kiely et al.[24], which is described briefly here. PM$_{2.5}$ concentrations have been simulated by WRF-chemv3.7.1, run at 30 km resolution with 33 vertical levels between the surface and 50 hPa. The simulation was run for August–October each year, after a 14 day spin up for chemistry. Meteorology was reinitialised every 15–16 days using National Centre Environmental Prediction Global Forecast System[90], with the meteorology free running between. Fire emissions are represented by FINNpeatSM, anthropogenic emissions are from EDGAR-HTAP2[91] for 2010 and biogenic emissions are from MEGAN (Model of Emissions of Gases and Aerosols from Nature)[92]. Gas-phase reactions were calculated by the MOZART (Model for Ozone and Related Chemical Tracers, version 4)[93] chemistry scheme and aerosol processes, binned into 0.039–0.156, 0.156–0.625, 0.625–2.5, and 2.5–10 μm, were represented by MOSAIC (Model for Simulating Aerosol Interactions and Chemistry)[94,95]. Secondary organic aerosol (SOA) formation from fires in the model is calculated as 4% of the fire emitted CO based on Spracklen et al.[96]. The contribution of fires to PM concentrations is calculated as the difference between simulations with and without fire. These simulations are the same as those from Kiely et al.[24], except for 2015 which in Kiely et al. was run with meteorology reinitialised once every month. This causes some differences to the simulated PM and consequently health impact estimates for 2015.

The population-weighted PM$_{2.5}$ (PW) is calculated using population data from the Gridded Population of the World, Version 4 (GPWv4)[97].

$$PW = \sum C_i * P_i / P_{tot} \quad (1)$$

where $C_i$ is the PM$_{2.5}$ concentration in a grid cell, $P_i$ is the population of a grid cell and $P_{tot}$ is the total population of the area.

Premature mortality per year, M, from disease j in grid cell i was calculated as,

$$M_{ij} = P_i I_j (RR_{jc} - 1) / RR_{jc} \quad (2)$$

where $P_i$ is the population in i, $I_j$ is the baseline mortality rate (deaths year$^{-1}$) for j, and $RR_{jc}$ is the relative risk for j at PM$_{2.5}$ concentration, c (μg m$^{-3}$). The PM$_{2.5}$ concentration is an annual average, and the average PM$_{2.5}$ from August from the simulation with no fires has been used to represent January to July and November to December. The baseline mortality rates and the population age composition are from the GBD2017[98], and the relative risks are taken from the Global Exposure Mortality Model (GEMM)[99] for non–accidental mortality (non-communicable disease and lower respiratory infections). The DALYs have been calculated as

$$DALY = YLL + YLD \quad (3)$$

where,

$$YLD_{ij} = P_i I_{YLL}(RR_{jc} - 1)/RR_{jc} \quad (4)$$

and

$$YLD_{ij} = P_i I_{YLD}(RR_{jc} - 1)/RR_{jc} \quad (5)$$

where $P_i$ is the population in i, $I_{YLL}$ and $I_{YLD}$ are the corresponding Years of Life Lost and Years Lived with Disability baseline rate (deaths year taken from GBD2017. $RR_{jc}$ is the relative risk from disease j at PM$_{2.5}$ concentration, c (μg m$^{-3}$), from the GEMM. The GEMM is validated against large cohort data which provides upper and lower uncertainty intervals, which have been used to create health estimates with a 95% uncertainty interval.

Using relative risk functions at different PM$_{2.5}$ concentrations is an established method for estimating the health impacts of PM$_{2.5}$ exposure[99], and has been applied previously to Indonesian fire emissions[25,26,100]. Using the GEMM and simulated PM$_{2.5}$ concentrations, estimated premature mortality resulting from fires in 2015 is 56,000, lower than previous estimates (76,000–100,000)[25,26] due to a less sensitive relative risk function[24].

**Cost of fires**. To calculate the economic cost of fire due to damages to agriculture and other land uses, we used the locations of fires combined with land-cover data to calculate the area of each land cover that was burned, which was then multiplied by the value of that land cover. We used FINNpeatSM to provide the locations of fires at 1 km$^2$ resolution. To account for heterogeneity in fire damage at smaller spatial scales, we scaled the area burned estimated by FINNpeatSM for Sumatra and Kalimantan in 2015 (63,938 km$^2$) by 0.59 to match the area burned estimated from analysis of Sentinel-1 (37,860 km$^2$), as found by Lohberger et al.[43]. We scaled FINNpeatSM burned area in other years by the same factor.

We identified the locations of oil palm plantations, wood fibre plantations, rubber plantations, crops, logging concessions and natural forest. The locations of oil palm, wood fibre, and rubber plantations come from the tree-plantations data for 2014, created by Transparent World, accessed from the Global Forest Watch. We used cropland categories from the European Space Agency Climate Change Initiative (ESA CCI) land cover[101] for 2015, downloaded from Global Forest Watch, with the oil palm plantation area from the tree-plantations data removed. The logging concessions data are for 2019 from the Ministry of Environment and Forestry. We identified natural forest as the primary forest categories in the ESA land-cover data, with the logging concessions removed.

There are uncertainties in the spatial distribution of land use. Combining land-cover categories from different datasets may result in some discrepancies. Dates of

the land-cover data vary and it is possible that some land-use types may have been established after the occurrence of fire. Remote sensing of land-cover types is also uncertain, for example the tree-plantations data claims an overall accuracy of 79%[102]. Data availability restricts us to static land-use / land-cover data. This necessary simplification means we will not represent longer term trends driven by changes in land use and land cover.

We estimated the value of land as the net present value (NPV) of that land use. We calculate financial impacts for the same year (2015) and adjust all values to 2015 US$. For forests we apply a NPV of $4079 ha$^{-1}$ based on 96 studies[103] placing a monetary value on provisioning, regulating and cultural services. We excluded the climate regulation value from the estimate as we estimate this separately. For all other land uses we apply average (across the rotation cycle of plantations) NPV from previous studies: oil palm ($8885 ha$^{-1}$)[104–110], rubber (US$1662 ha$^{-1}$)[105,106], crops (rice and maize; $827 ha$^{-1}$)[105], logging (US$7713 ha$^{-1}$)[105,107,108], and wood fibre (Acacia; US$1206 ha$^{-1}$)[105].

To calculate the imputed damage value of $CO_2$ emissions, we combined $CO_2$ emissions from the FINNpeatSM emissions inventory[24] by the average 2009–2020 closing price of $CO_2$ in the EU ETS[111] (€10.8 t$CO_2^{-1}$) converted to US$ using an exchange rate of 1.09 to give US$11.8 t$CO_2^{-1}$). This is similar to the US$10 t$CO_2^{-1}$ used in other studies[112,113].

To calculate the economic cost of the health impacts of fires, we multiplied the number of the disability adjusted life years (DALYS) due to smoke exposure by the economic value of a DALY. To estimate the economic value of a DALY we used the economic loss due to non-communicable diseases (NCDs) in Indonesia from 2012 through 2030[114], estimated as US$4.47 trillion which equates to US$235 billon yr$^{-1}$. We assume 50 million DALYS per year from NCD[115] to calculate a cost per DALY of US$4710. Compared with welfare-based and income-based methods for estimating the cost of air pollution, as described in the World Bank and Institute for Health Metrics and Evaluation report[116], our method allows us to consider the cost from all health impacts of fires, rather than mortality only. It also uses data specific for Indonesia, whereas a welfare-based method would require adjusting from studies in other countries.

We apply the same economic value to DALYs in all countries, so the cost to Malaysia and Singapore may be underestimated. While the other costs estimated in this study are from Indonesian fires only, the simulations which the DALYs are calculated from also include some fires in Malaysia, Brunei and Thailand. These non-Indonesian fires contribute only 3–7% of the $PM_{2.5}$ emissions in different years. For 2015, simulated $PM_{2.5}$ from non-Indonesian fires only has been used to estimate that these fires cause 3% of the mortalities and DALYs from fires, and in other years this is likely to be similar (see supplement).

For this study we are comparing fire events in different years, rather than considering a period of time. We therefore keep NPV, $CO_2$ and DALY costs constant for each year so that the only difference between costs in different years are due to differences in fires. This means the costs are relative only to the magnitude of each fire event, rather than to when the event occurred.

The reduced costs of using fire to clear land compared to more expensive mechanical methods has also been calculated. The difference in cost of fire and zero-burning clearing methods have been taken from Guyon and Simorangkir[45] as US$156 ha$^{-1}$ for non-peatland and US$848 ha$^{-1}$ for peatland, which is an average for 2004–2015 after adjusting for inflation. This total reduction in costs is calculated as the area of fire on primary forest, assuming that this was all intentionally burnt for land conversion, multiplied by the reduced cost per hectare. We assume no difference in land clearance costs for other land covers[45,45]. The reduced costs of using fire as a land clearing method vary depending on the land and region, and we have used the upper estimate where multiple values are given.

**Peatland restoration.** To estimate the potential impacts of peatland restoration, we assumed that peatland areas could be restored to the conditions currently found within protected areas. To determine the effects of protecting land, we analysed the burned area and soil moisture inside and outside of protected areas. We used the peatland distribution map from the World Resources Institute to determine peatland extent. The soil moisture is the Soil Moisture Active Passive (SMAP) product from NASA[117] at 9 km resolution. The protected area data comes from the World Database on Protected Areas, downloaded from the Global Forest Watch. The protected areas are split into nine categories; Game Reserve, Grand Forest Park, Hunting Park, National Park, Nature Recreation Park, Nature Reserve, Protection Forest, Wildlife Reserve, and Undesignated. We have included World Heritage Parks, Ramsar Wetlands of International Importance and UNESCO biosphere reserves as National Parks. For each protected area the August–October peatland burned area per km$^2$ inside each protected area was compared with the August–October peatland burned area per km$^2$ within a 0.25° boundary of the protected area. The ratio of these two values was calculated for each protected area, for each year. The average peatland soil moisture inside each protected area for each month in August–October 2015 was compared with the average peatland soil moisture within the 0.25° boundary of the protected area, giving a soil moisture ratio for each protected area for each month.

The average ratio of peatland and non-peatland burned area and soil moisture was found across all protected areas in each category, for Sumatra and Kalimantan separately (Supplementary Table 2). We used the average burned areas and soil

moisture ratios from National Parks in Kalimantan to estimate the area burned and emissions after restoration. The FINNpeatSM emissions were re-calculated with the burned area and soil moisture in restored areas scaled by these ratios. Where the restored areas are only partially peatland, fires not on peatland are scaled by the non-peatland ratio for burned area (Supplementary Table 2).

The areas to be restored were selected by finding the 0.2 × 0.2° (48 800 ha, 488 km$^2$) gridcells with the greatest total dry season emissions between 2004 and 2015. Only gridcells containing at least 25% peatland were considered for restoration. For the case when 2.49 Mha of land is restored, 2.25 Mha of this is peatland. Smaller gridcells at 0.1 × 0.1° (122 km$^2$) were also considered; however, evaluation of the benefits of protected areas based on size suggests that larger protected areas have greater fire reduction than smaller protected areas (see supplement). Emissions were also calculated when areas for restoration were randomly placed on peatland in Sumatra and Kalimantan. To get a random allocation of gridcells the random python module was used, which uses the Mersenne Twister pseudorandom number generator. This random allocation was repeated 10 times, and the average emissions across all these scenarios calculated. The range between the 10 scenarios is small (<3%). To evaluate the emissions reduction as the size of restoration increases, the number of 488 km$^2$ cells restored has been increased in intervals of 5 up to 100, and then in intervals of 50 up to 500, with new fire emissions created for each case.

To calculate the cost of peatland restoration an estimated cost of canal blocking was combined with the buy-back cost of land to be restored. Oil palm in Riau (Sumatra) that is ready to harvest sells for US$3077 ha$^{-1}$[118]. The World Bank[22], suggests a greater onetime buy-back cost for oil palm of US$10,000 ha$^{-1}$. These two values have been used as lower and upper estimate of the buy-back cost. We assume a buy-back cost of other land uses based on the NPV: US$418—1357 ha$^{-1}$ for wood fibre and US$287—931 ha$^{-1}$ for cropland.

**Health impacts after restoration.** For 2015, we simulated $PM_{2.5}$ concentrations using the WRF-chem model with emissions from the peatland restoration scenario. We then recalculated health impacts calculations using these simulations. The impact of restoration on public health is estimated as the difference in health impacts between the baseline simulation and the simulation with emissions from the restoration scenario.

For years other than 2015, we estimate the reduction in DALYs from restoration directly from the reduction in $PM_{2.5}$ emissions. The DALYs from exposure to smoke from fires decrease linearly with the $PM_{2.5}$ emissions, at a rate of 0.16 million DALY Tg$^{-1}$ $PM_{2.5}$ (see supplement). We use this relationship to estimate the DALYs under the restoration scenarios, as was done for premature mortality in Kiely et al.[24]. For 2015, using $PM_{2.5}$ emissions and the linear relationship results in post-restoration DALYs within 1% of those estimated using exposure to simulated $PM_{2.5}$ concentrations simulated by WRF-chem, demonstrating that this simple approach is sufficient to estimate DALYS.

**Reporting summary.** Further information on research design is available in the Nature Research Reporting Summary linked to this article.

## Data availability

The fire emissions and burned area data used in and generated for this study have been deposited in the NERC EDS Environmental Information Data Centre and can be accessed at https://doi.org/10.5285/fdae44ed-8b22-4935-b889-b4b271138385. The Transparent World tree-plantations data used in this study can be accessed from Global Forest Watch at https://data.globalforestwatch.org/datasets/gfw::tree-plantations/about. The European Space Agency land-cover CCI data used in this study can be accessed from www.esa-landcover-cci.org/?q=node/164. The logging concessions data from the Ministry of Environment and Forestry Indonesia used in this study can be accessed from http://geoportal.menlhk.go.id/arcgis/rest/services/KLHK. The World Resources Institute Peat lands data used in this study can be accessed from Global Forest Watch at https://data.globalforestwatch.org/datasets/gfw::indonesia-peat-lands/about. The World Database on Protected Areas (WDPA) from the United Nations Environment Programme (UNEP) and the International Union for Conservation of Nature (IUCN) can be accessed from Protected Planet at www.protectedplanet.net/en.

## Code availability

All codes produced for this study used Python v2.7, and are available from the authors upon reasonable request.

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

## Acknowledgements

L.K. was funded by a studentship from the NERC SPHERES Doctoral Training Partnership (NE/L002574/1) and by the United Bank of Carbon (UBoC). We acknowledge the use of the WRFotron scripts developed by Christoph Knote to automatise WRF-chem runs with re-initialised meteorology.

## Author contributions

Conceptualisation by L.K., D.V.S., and S.R.A. Modelling work and analysis was completed by L.K., advised by D.V.S., S.R.A., and E.P. Health impact calculations were done by L.C. and L.K. Analysis of protected areas was completed by L.K. with help from H.A.A. C.K. provided WRFotron modelling scripts. L.K. provided fire emissions with help from C.W. L.K. prepared the manuscript, assisted by all authors.

## Competing interests

The authors declare no competing interests.
