## [Peer Review File · Nature Communications]

Assessing economic cost of Indonesian fires and the benefits of restoring peatlandREVIEWER COMMENTS

Reviewer #1 (Remarks to the Author):

General Comments:

This paper evaluates the economic costs of fires in Indonesia and benefits of peatland restoration strategies. This expands on preliminary reports of the economic costs of fires that have been less comprehensive or have looked at individual years. Overall, this is an interesting paper that adds a new dimension to these prior studies. I would like to see a few improvements in the manuscript. This includes: 1) a more comprehensive introduction that provides more connections to prior work, 2) a more in depth description of uncertainties in the analysis, and 3) a possible reorganization of some sections of the paper. Please see more detailed comments below.

Abstract:

-Line 20: Do you have an estimate of the costs for completing this restoration activities? This would be helpful for the later statement of cost effectiveness.

Intro:

-Line 29: This paragraph needs a little bit more detail to better motivate the paper. Can you help put Indonesia's fires in the context of global fire emissions and highlight some of the most intense fire seasons?

-Line 33: Some quantitative estimates here would be helpful.

-Line 37: Provide more regional context on fires and damages for readers who may not be familiar.

-Line 40: Please help clarify this comparison with Amazonian fires, since the latter study mentions additional categories for damage impacts. Was this year of comparable severity in both countries?

-Line 48: Provide date of moratorium and indication of effectiveness of these efforts (if available).

-Overall I think that the intro would benefit from a more thorough review of the literature to help put this study in context. Also, there are some studies that are in the results section that could perhaps be moved here.

Results:

-I found the ordering of the subheadings in this section a little difficult to follow.

-Lines 64-70: This material might be more useful in the introduction or a discussion section.

-Line 75: How is the rotation cycle of plantations taken into account here?

-Figure 1: Could you add a second y-axis with a line showing fire emissions?

-Line 113: Similar to comment above. Are these estimates for clearing natural vegetation for agriculture or at the end of the harvest cycle, or a combination?

-Line 157: Can you link this reduction to emissions as described earlier in the paragraph? It's unclear how the change in soil moisture is used in the rest of the analysis.

-Line 168: How was this restoration area estimated?

-Figure 3: It's hard to see most of the detail in any of the panels except for (d) and (e). Is there a different way to make these maps to make more clear?

-Line 185: Total burned area across Indonesia? Or across Sumatra and Kalimantan? And within peatlands only, or fires in non-peatland areas?

-Line 220: Figure 4 and figure 2 show the importance of inter annual variability. Can you comment on how having some static land use / land cover data for single years might influence these results?

-Line 253: If the goal is to prioritize restoration, how does the emissions magnitude change in locations that have burned multiple times? Do we expect that areas that have the highest emissions in the past would have high future emissions, or would the fuel loads be much lower?

Conclusions:

-Line 297: Perhaps rename the policy implications section to discussion or conclusions, as it seems that the goal of this section is to comment on the broader implications of the work, but it does not present new results.

-Line 303: How does this fit with previous work that looks at the actors responsible for starting fires? If a fire is detected in a specific area, it may or may not have originated there.

Methods:

-Line 342: Can you provide some figures of the fire data in the supplement that the reader can use for reference? For readers not familiar with the paper that is mentioned, how much did this correction factor change the results from the original FINN emissions?

-Line 354: Do you expect the scaling factor to hold regardless of the severity of the fire season from year to year?

-Line 367: Can you provide some of the dates that were used? Did they vary over time or is each dataset a snapshot of land use/land cover?

-Line 370: I would like to see a further description of how this uncertainty fits into other sources of uncertainty for the economic calculations.

-Line 426: If the DALY's estimated here are then used in the economic calculation, shouldn't this section be presented first?

-Line 465: Has the model been validated against any observations to provide an estimate of uncertainty?

-Line 470: Have the protected areas in this analysis been converted to different land uses before? Do you expect the recovery of previously degraded peatlands to match more primary forested areas? If not, how would this affect uncertainty in the analysis?

Supplement:

-Table S2: What is the land use history of these different protected areas? Have some areas been converted and are now recovering under protected status? Can you also comment in the text on differences in the ratio and potential efficacy of these protected areas under drought vs. non-drought years in reducing fire activity?

-Can you provide some further details on the designation/management practices between grand forest parks, national parks, nature reserves, etc.?

Reviewer #2 (Remarks to the Author):

In this study the economic impact of peatland restoration is presented for Indonesian peatlands, following a number of possible scenarios where currently committed goals for peatland restoration had already been met.

This is a timely analysis as Indonesia's Peatland Restoration Agency is currently struggling to meet its restoration targets and appeared to be under resourced given the scale of the problem. The economic cost of failing to meet these targets are not necessarily obvious and it is great that these are being calculated and well presented as the authors have done. The hypothetical what-if scenario presented here is an elegant and scientifically valid way of demonstrating the costs that have been born through past inaction.

While I support publication of this study, I would like to see better discussion of the assumptions contained within, none of which are necessarily unreasonable, but which I think could be better justified. I also have a number of smaller suggestions which I think could improve the manuscript.

Main points:

- The authors argue for targeted restoration towards previously burnt sites, yet at no point was evidence provided to show that previously burnt sites are more likely to burn in subsequent years. The likelihood of repeat burning is something that needs to be established early on in the

manuscript. Although I am aware this is often the case, I think it requires some support from the literature.

- It is not yet fully clear what effect peatland restoration will have on fire prevention. I think in the short term, restoration is not likely to be so effective as to reduce fires in restored peatlands to the same degree as that of a natural system (as the authors have assumed), especially given that restored sites have been found to still have lower surface moisture contents compared to natural peatlands (ref 33, 34). However, given the current lack of data, I think it is acceptable to make this assumption, providing it is discussed and justified more than it is presently. I think it would suffice to briefly summarise what is and is not known about the effect of restoration on fire suppression, both in terms of burn intervals and fire severity.

- I don't fully agree with the statement on line 223 where the authors state their analysis to be conservative in terms of burn scenarios, if restoration fails to reduce burn events to that of a natural system, which is entirely reasonable, then the analysis is not conservative. I suggest this is better justified or deleted

Minor points:

- Given that fires are still occurring in protected peatlands and national parks, I can see why the no-burn scenario is presented as a less likely scenario. However, I think it is worth stressing that a no-burn scenario was the default state for the majority of Indonesia's tropical peatlands prior to the last few hundred years. This is especially relevant if we are to use the present day burn rate in national parks as a 'pristine' reference point for restored sites. This paper may be useful <https://doi.org/10.1016/j.quascirev.2014.05.002>

- Figure 2: Do the authors mean 0.25 degrees latitude and longitude? (or both?). A legend for Kalimantan and Sumatra would improve this figure, as would annotating the plot to specify the blue percentage is reduction in burned area. At the moment there is rather a lot going on in this plot and the clarity would be improved by moving some information from the caption into the plot itself. The thick black line at zero would look better as a faint or dotted line, also the mean value for all protected areas in Kalimantan in 2015 looks to be exactly 1 which seems odd, the authors might want to check this data point.

- Figure 5: Please define DALYS in the caption

Reviewer #3 (Remarks to the Author):

Key results

The results presented in this paper are timely and highly relevant. The work demonstrates that taking action to prevent peatland forest degradation will have very positive impacts on both human health and carbon emissions resulting from anthropogenic activity. It provides a scientific basis to support land management policy in Indonesia. With a focus on the impact of wildfires and land clearing on peatland forests, this work also has relevance to other parts of the world, including north America and Canada.

One weakness in the manuscript that I find concerning is the lack of emphasis put on loss of biodiversity and local subsistence livelihoods which are also significantly impacted by such fires. While it is understood that the analysis concentrates on 3 dimensions of fire impacts (land cover, carbon emissions and human health), I consider it vital that more statements should be included to ensure that the impacts on biodiversity and subsistence livelihoods are not neglected in the conclusions. Policy makers must not be led to believe that such impacts are trivial, even though it may be very difficult to assess them.

Validity

Notwithstanding the comments above, the data and analysis presented in this manuscript appears to present a valid statement about the geophysical impact of fires in Kalimantan and Sumatra. I am a little concerned however about the robustness of the financial and socioeconomic impacts which are based on less reliable data and tend to rely on more sweeping assumptions. For example, between lines 72 to 76, and other places in the manuscript, and supplementary information, the 'costs' of fire damage are based on NPVs of each land use, sourced from a very limited number of publications which are some years old. It is not clear if the changes in price for the agricultural products mentioned over the period of years covered are taken into account. Also, a single value of forest seems to be used (supp info table S1). Furthermore, the conversion of these cited values to US\$ at the time of their publication perhaps do not represent the current value. In addition, these impact values are based on market prices and therefore do not include the consumer/producer surplus components of what makes up full economic value. To give more confidence in these financial estimates, the authors need to make it clear how these issues are dealt with.

In figure 1 of the manuscript for example, the title suggests that these are factual values, while in reality they are estimates, and it should say that. In that same figure, it differentiates health costs from Indonesia, from those in Singapore and Malaysia, and while that is important, it is not clear how those values from other countries are included in the total estimates for Indonesia (the focus of the paper). While this is discussed in the SI, this could be made clearer in the conclusions of the paper. (This could be an advantage from the study in that it provides insights into the problem for the neighbouring countries)

In relation to the health impacts, the use of Dalys from Non Communicable Diseases is of importance, although it must be noted that not all NCDs are the result of fires (eg diabetes and many forms of cancers). Perhaps this should be clarified in the text. Also, I would like to see reference to other publications rather than a total reliance on Keily et al for this part of the analysis. For example, the manuscript could benefit from citing other work such as:

- Proc Natl Acad Sci 115(38):9592–9597. <https://doi.org/10.1073/pnas.1803222115>
- Environmental Science and Pollution Research volume 26, pages 31315–31327 (2019)
- Sci Rep 6:37074. <https://doi.org/10.1038/srep37074>
- Shannon N Koplitz et al 2016 Environ. Res. Lett. 11 094023

Furthermore, while Dalys are useful, they do only account for health burden rather than including a wider assessment of public health costs. No mention is made of the costs associated with short and long term impacts on mental health.

Throughout the manuscript there seems to be some confusion between the terminology around costs, losses and values. This should be clarified. For example in lines 18/19, and elsewhere, the term 'costs' could be better replaced by 'economic losses'. On line 80, the word 'costs' could be replaced by the term 'imputed damage value of'. While some may think these points are trivial, they do have a significant meaning within the discipline of economics. Some further discussion on the section on costs of fires is provided below under methodology.

Significance

It is my opinion that while I have outlined some weaknesses in the way the information is provided in the manuscript, I believe that this paper is an important addition to the relevant literature. It highlights the importance and benefits of large-scale landscape remediation and the means by which individual countries can contribute to reversing global degradation. I think it makes use of important scientific findings to support essential land use policy change, in particular the importance of developing and protecting national parks.

Data and methodology

The approach used in this manuscript takes advantage of some of the most recent advances in earth observation data to quantify the extent of fires in Indonesia. It combines this with an interesting method of evaluating the potential extent of fires in restored peatland areas. While the use of fire incidence and extent (and soil moisture levels) in protected areas may be a proxy for those that may occur in restored peatlands, the authors should clarify exactly how much of the protected areas included in the analysis are actually 100% peatland, as the information in the text

and on Table S2 only specifies the non-peatland areas of national parks, but does not explicitly state that the other types of protected areas are actually all peatland areas. (ie could be more explicit about this in section 2 of SI). When considering the effectiveness of Peatland restoration, the text (lines 173/174) suggests that data on soil moisture and burned area applied to all 51 hypothetical restoration areas is only drawn from Kalimantan, but these sites are distributed between Kalimantan and Sumatra, but conditions and impacts of restoration in the latter are different, as illustrated by Figure 2. Perhaps this should be clarified/justified if no other data from Sumatra is available.

While the use of the 'no fire' and 'no peatlands fire' scenarios are useful to investigate model uncertainty, it must be noted that there are many costly and practical difficulties in achieving the 'no peatlands fire' option in drought periods as large-scale rewetting (line 182) is not likely to be possible.

I am not an atmospheric scientist nor modeller and so am not fully qualified to evaluate the manuscript values on levels of CO₂ or particulate matter. I would however like to see more explicitly how these estimated impacts effect populations both near to or more distant from restoration sites (most of these likely being some distance from large human populations). Perhaps some kind of gradient figure could be provided since gridded population data is used.

The maps shown in figure 3 are rather small and the similar colour palettes make it quite difficult to understand the information being presented therein. For example, is no account taken of potential DALYs occurring in Java as a result of these fires?

In terms of the section on scaling up of restoration efforts, while this is an important section, it is highly unrealistic to ignore the important socioeconomic and institutional factors which would influence the success of such a policy (especially in Indonesia). For example, land ownership and cooperation from land users would be essential if such scaling up would be achieved. While this is briefly mentioned in the policy implications section (lines 306/307), I feel that it should be included more explicitly in this scaling up section, and perhaps repeated in the conclusions.

In the section on costs of fires, I have a number of concerns, both in terms of terminology/semantics and in terms of theoretical underpinning. For example, although the authors have discussed the issue of heterogeneity of fire damage, this has tended to be based on the geographical scale of fire areas rather than the heterogeneity within land cover types. Of particular concern is the lack of recognition of the imputed value of subsistence agriculture which often will be occurring within or on the margins of other commercial land uses, but is very rarely reported in any data or official reporting.

The value of land cover damages as presented in table 1 of the SI and in the text (lines 372 to 381) are worryingly dependent on just two rather old references (refs 51 and 52) although a massive literature on this is available. I suggest some more references on land use values could be found. Again, the value of forest to local forest users and subsistence farmers is significantly underplayed in this manuscript (limited to pest control and pollination – mostly relevant to commercial farmers as these issues are mainly taken care of by the large number of plant types included in traditional tropical forest subsistence farms, eg in Amazonia, Vanuatu, Malaysia etc as well as in Indonesia).

No attempt is made to include any costs associated with rewetting, although this is suggested as an option.

In the section in lines 414 to 423, I find the very idea that there is any economic benefit from burning any forests, in particular tropical forests, is contrary to prevailing views. Since the loss of biodiversity, habitats and endangered species are amongst the highest in these forest areas, it is surely unwise to be suggesting that there is some level of economic benefit from burning them. Indeed the reference used (no 26) has been written from the perspective of fire as a tool for agriculture. It is wrong to use this to suggest that large-scale burning of forests is in any way a benefit to anyone or anything, except for the bottom line in unscrupulous large-scale forest and agribusinesses.

Why do the authors assume (eg on line 420) that there are any economic benefits from forest clearing, especially since they have failed to take account of biodiversity values, option values of potential pharmaceuticals etc. This point also questions the validity of the authors use of the term

'cost benefit analysis' (line 319 and ref 28), which cannot be correctly applied in this situation either practically (not enough available reliable values for all components of peatland forests) or theoretically (no clear mechanism by which all the considerations of welfare economics can be brought to bear on this problem. Maybe the term cost effectiveness analysis would be better as a true cost benefit analysis should highlight the processes by which the losers can be compensated by the winners from such calculations. While it is clearly of value to mention land buybacks (lines 277, 280), the lack of inclusion of biodiversity and subsistence values in this would render it inadequate or unworkable.

Other points relating to the cost sections include:

- Has the changing value of the US\$ exchange rate been taken into account in different years (given land use values would be in Indonesian rupiah and then converted)?
- It appears that there has been no attempt to include any estimate of the impact of fires on employment (+ or-)
- Have the public health costs of treating health impacts of fires been included?

Analytical approach

I believe I have covered this above.

Suggested improvements

- A very clear statement should be made at the start of the paper that not all costs and impacts of fires are included
- Could the title be improved? The paper does not include all the economic costs of Indonesian fires – perhaps it would be simpler and more accurate to have the title as: 'Assessing costs of Indonesian fires and benefits of restoring peatland'

Clarity and context

- Clarification of terminology – eg costs vs losses (line 18/19 and elsewhere)
- Need for greater transparency about what is not included in the analysis
- Would it not be better to put the policy implications and conclusions sections after the methods?

References

In my opinion, more up to date references relating to costs of fires, values of forests and other land uses etc could be used.

Too much reliance on methodology of Keily et al in fire emissions inventory – ref to other approaches should be provided eg:

- Battye, W. and Battye, R (2002) Development of Emissions Inventory Methods for Wildland Fire Final Report February 2002. US EPA.
- van der Werf, G. R., Randerson, J. T., Giglio, L., van Leeuwen, T. T., Chen, Y., Rogers, B. M., Mu, M., van Marle, M. J. E., Morton, D. C., Collatz, G. J., Yokelson, R. J., and Kasibhatla, P. S.: Global fire emissions estimates during 1997–2016, *Earth Syst. Sci. Data*, 9, 697–720, <https://doi.org/10.5194/essd-9-697-2017>, 2017.
- Walker, X.J., Baltzer, J.L., Cumming, S.G. et al. Increasing wildfires threaten historic carbon sink of boreal forest soils. *Nature* 572, 520–523 (2019). <https://doi.org/10.1038/s41586-019-1474-y>

My expertise

I am not able to fully evaluate the climatological and regional chemical modelling included here.

Professor Caroline A Sullivan, NSW Australia.

Assessing costs of Indonesian fires and the benefits of restoring peatland Kiely et al.

Response to Review

We thank the reviewers for their comments on our work and for their thoughtful and constructive review. We respond to the reviewer comments (in black text) below. Our responses are in blue text. We have been able to respond to all the reviewer comments and have revised our manuscript accordingly. We feel that our revised manuscript is much improved.

REVIEWER COMMENTS

Reviewer #1 (Remarks to the Author):

General Comments:

This paper evaluates the economic costs of fires in Indonesia and benefits of peatland restoration strategies. This expands on preliminary reports of the economic costs of fires that have been less comprehensive or have looked at individual years. Overall, this is an interesting paper that adds a new dimension to these prior studies. I would like to see a few improvements in the manuscript. This includes: 1) a more comprehensive introduction that provides more connections to prior work, 2) a more in depth description of uncertainties in the analysis, and 3) a possible reorganization of some sections of the paper.

We have modified our manuscript as requested by the reviewer.

1) We have extended our introduction and discussion to provide more details and connections to previous work. We provide specific details in the responses below.

2) We have added an additional description and discussion on the uncertainties. We add the following text:

“There are a range of uncertainties in our analysis. Estimates of the losses and damage to crops, plantation and forest conflate uncertainties in burned area, uncertainties in land use mapping and the value of different land covers. Estimates of economic costs of CO₂ combine uncertainties in burned area, biomass consumption linked to vegetation loads and burn depth and emission factors⁶⁷, as well as the damage value associated with CO₂. Emission estimates do not account for declining emissions from recurrent fires on drained peatlands⁵⁶. Health impact estimates combine uncertainty in exposure to particulate matter with uncertainty in concentration response functions and economic costs of health impacts. Despite this large range of uncertainties our estimates are consistent with previous estimates of emissions^{23,24,68}, population exposure to particulate matter⁶⁹⁻⁷¹, associated health impacts^{24,26,71} and economic costs²². ”

Detailed responses to point by point comments from the reviewer are below.

Please see more detailed comments below.

Abstract:

-Line 20: Do you have an estimate of the costs for completing this restoration activities? This would be helpful for the later statement of cost effectiveness.

We have estimated the cost of restoration in the paper, finding it to be between US\$3.7 billion and US\$7 billion. This estimate has been added to the abstract as follows:

'In order to reduce fire, the Indonesian government has committed to restore 2.49 Mha of degraded peatland, with an estimated cost of US\$3.2-7 billion.'

A previous study (Hansson & Dargusch, 2018) estimated the cost of restoration as \$4.6 billion and is consistent with our estimate. We add:

"Another study estimates that restoration costs would exceed US \$4.6 billion⁴⁸"

Intro:

-Line 29: This paragraph needs a little bit more detail to better motivate the paper. Can you help put Indonesia's fires in the context of global fire emissions and highlight some of the most intense fire seasons?

To provide a bit more detail we added information on the years with the biggest fires and placed Indonesian fires in context with global fire emissions. We added the following:

'with the two largest fire events on record occurring in 1997 and 2015^{2,3}.'

And:

'Fires in Equatorial Asia (mostly Indonesia) were responsible for 8% of global fire carbon emissions in 1997-2016²³.'

-Line 33: Some quantitative estimates here would be helpful.

Quantitative estimates from the literature have been added:

'Indonesian peatlands store an estimated 57 Gt of carbon, 55% of the world's tropical peatland carbon⁷⁸'

'Peatland fires cause large CO₂ emissions¹⁹⁻²¹, contributing substantially to Indonesia's greenhouse gas emissions²². Fires in Equatorial Asia (mostly Indonesia) were responsible for 8% of global fire carbon emissions in 1997-2016²³'

-Line 37: Provide more regional context on fires and damages for readers who may not be familiar.

We add a statement about fire damage in other regions for context:

"The losses and damages caused by landscape fires cost billions of US\$ and exceed 1% of gross domestic product (GDP) in countries where fire is prevalent²⁸."

-Line 40: Please help clarify this comparison with Amazonian fires, since the latter study mentions additional categories for damage impacts. Was this year of comparable severity in both countries?

The size of the Amazon and Indonesian fire events was comparable both in terms of area burned and the losses and damages incurred. The area burned in the Amazon in 1998 fires has been included to clarify this comparison.

-Line 48: Provide date of moratorium and indication of effectiveness of these efforts (if available).

Dates for the moratorium and announcement of peatland restoration have been added to the text, as has the following sentence on the effectiveness of the moratorium:
'Recent studies have found that the moratorium on land conversion may not have been effective in reducing deforestation or fires²⁴²⁵.'

-Overall I think that the intro would benefit from a more thorough review of the literature to help put this study in context. Also, there are some studies that are in the results section that could perhaps be moved here.

As suggested, we have improved the introduction to provide a more thorough review of the literature. We have extended the introduction to provide context on peatlands in Southeast Asia (extent, C storage and other benefits), the causes and impacts of peatland degradation and the issues around restoration. We better motivate our study through highlighting the knowledge gaps and clearly stating the aim of our paper.

Results:

-I found the ordering of the subheadings in this section a little difficult to follow. The subheadings have now been edited and are hopefully clearer.

-Lines 64-70: This material might be more useful in the introduction or a discussion section. The description of the sectors included in the World Bank cost estimate is in the introduction already. It has now been removed from this paragraph to avoid repetition.

-Line 75: How is the rotation cycle of plantations taken into account here? The rotation cycle of plantations is accounted for within our calculation of the net-present value of plantations. We add a statement to the methods to clarify.

-Figure 1: Could you add a second y-axis with a line showing fire emissions? The fire PM_{2.5} emissions have been added to this plot.

-Line 113: Similar to comment above. Are these estimates for clearing natural vegetation for agriculture or at the end of the harvest cycle, or a combination? This is an estimate of the cost of clearing natural vegetation for development of agriculture.

-Line 157: Can you link this reduction to emissions as described earlier in the paragraph? It's unclear how the change in soil moisture is used in the rest of the analysis. The following has been added to make the link between soil moisture and fire emissions clearer:
'Greater soil moisture is likely to reduce the burn depth of fires, resulting in lower emissions.'

-Line 168: How was this restoration area estimated? We have given some more explanation in the text here, and directed to where the approach is explained in more detail:
'We assumed restoration areas of ~500 km² based on the evaluation of protected areas (see methods and table S4)'

-Figure 3: It's hard to see most of the detail in any of the panels except for (d) and (e). Is there a different way to make these maps to make more clear?

We have deleted panels (b) and (c) which showed emissions as these map closely to burned area. The 4 panel plot showing locations of restoration, reduction in burned area, change in particulate pollution and change in health impacts is now much clearer.

-Line 185: Total burned area across Indonesia? Or across Sumatra and Kalimantan? And within peatlands only, or fires in non-peatland areas?

This is referring to total burned area across Sumatra and Kalimantan, for peatland and non-peatland. This has now been specified in the text:

'In 2015, 15% of the total burned area in Sumatra and Kalimantan occurred on areas selected for restoration'

-Line 220: Figure 4 and figure 2 show the importance of inter annual variability. Can you comment on how having some static land use / land cover data for single years might influence these results?

This short-term interannual variability is largely driven by climate. Changes in land use and land cover will contribute to longer term trends in fire, emissions and costs. We add a sentence to the Methods to acknowledge our use of static land use / land cover data:

'Data availability restricts us to static land use / land cover data. This necessary simplification means we will not represent longer term trends driven by changes in land use and land cover.'

-Line 253: If the goal is to prioritize restoration, how does the emissions magnitude change in locations that have burned multiple times? Do we expect that areas that have the highest emissions in the past would have high future emissions, or would the fuel loads be much lower?

This is a good point. Emissions from peatland fires decline with recurrent fires so the greatest emission reductions will be from restoration of unburned peat within areas of high fire risk. We add the following:

"Carbon emissions are greatest the first time a peatland burns and typically decline with subsequent fires⁵⁶. Restoring unburned peatlands in areas of high fire risk will therefore lead to the greatest reduction in emissions. At the scale of our analysis, areas designated for restoration will include both burned and unburned peatland."

The impact of previous burn history on emissions is not fully accounted for in existing emission inventories. We add the following line to our discussion of uncertainties:

"Emission estimates do not account for declining emissions from recurrent fires on drained peatlands⁵⁶"

Conclusions:

-Line 297: Perhaps rename the policy implications section to discussion or conclusions, as it seems that the goal of this section is to comment on the broader implications of the work, but it does not present new results.

This section has been renamed to *'Conclusions and implications for policy'*.

-Line 303: How does this fit with previous work that looks at the actors responsible for starting fires? If a fire is detected in a specific area, it may or may not have originated there. We agree that the starting location of a fire is important for determining who is responsible. Unfortunately, finding where a fire started is difficult, which is one factor hindering the effectiveness of fire bans. We add the following text to our discussion:
'Attribution of fire is difficult at a local scale due to fire spread, overlapping land claims, impacts of drainage on neighbouring land⁸⁰, complicating efforts to effectively direct fire suppression efforts.'

Methods:

-Line 342: Can you provide some figures of the fire data in the supplement that the reader can use for reference? For readers not familiar with the paper that is mentioned, how much did this correction factor change the results from the original FINN emissions?
We note that the new emissions scheme (FINNpeatSM) provides emissions from peat combustion in addition to the vegetation emissions provided by the standard (FINN) emissions scheme. We added a figure to the supplement showing emissions from FINNpeatSM and FINNv1.5 for 2004-2015, and the following has been added to the text:
'Total FINNpeatSM PM_{2.5} emissions in 2015 are a factor of 3.5 greater than those given by FINNv1.5⁵².'

-Line 354: Do you expect the scaling factor to hold regardless of the severity of the fire season from year to year?

This scaling factor is largely caused by the difference in resolution between high resolution Sentinel burned area being able to identify unburned patches of land within a large burn scar that can not be identified in our coarser resolution analysis. This difference in resolution is the same between years and so is likely to hold. However, we note Sentinel data is only available from 2015, so it is not possible to assess year to year variability.

-Line 367: Can you provide some of the dates that were used? Did they vary over time or is each dataset a snapshot of land use/land cover?

We have added the dates for the landcover datasets. Each dataset provides a snapshot of land use / land cover. We have added a sentence to the Methods to acknowledge this.

-Line 370: I would like to see a further description of how this uncertainty fits into other sources of uncertainty for the economic calculations.

It is difficult to provide a full quantitative estimate of uncertainty. We note that previous studies of the economic costs of fires have also not been able to provide this. We agree that a better description and discussion of uncertainty is needed and we have added a discussion of uncertainty to the paper.

-Line 426: If the DALY's estimated here are then used in the economic calculation, shouldn't this section be presented first?

We have swapped the order as suggested.

-Line 465: Has the model been validated against any observations to provide an estimate of uncertainty?

The GEMM is validated against large cohort data which provides upper and lower uncertainty intervals. These have been used to produce mortality estimates with a 95% uncertainty interval, as shown on Figure S1. This has been added to the methods: *'The GEMM is validated against large cohort data which provides upper and lower uncertainty intervals, which have been used to create health estimates with a 95% uncertainty interval.'*

-Line 470: Have the protected areas in this analysis been converted to different land uses before? Do you expect the recovery of previously degraded peatlands to match more primary forested areas? If not, how would this affect uncertainty in the analysis?

Protected areas have a diverse land-use history consisting of primary forest, recovering secondary forest and previously cleared land that is being restored. Protected areas also experience ongoing threats of deforestation and degradation. Illegal selective logging is common, which often involves digging drainage canals for transport, so the forests in protected areas may be partly drained. Analysis of deforestation rates in protected areas show variable effectiveness (Gaveau et al., 2013; Brun et al., 2015; Spracklen et al., 2015). Our analysis in the manuscript also shows that protected areas experience deforestation and fire. Protected areas therefore provide a useful target for restoration.

Currently there is very little data on the fire behaviour of restored peatland forests, but if it is much more fire prone than current protected forest then the emissions reduction will be less. We have added the following to line 230 make this clear:

'There is currently little data on the susceptibility of restored peatland to fire, and the effectiveness of peatland restoration interventions are limited to small-scale trials⁴⁸. If peatland restoration is not effective as we assume here, we will overestimate emission reductions.'

Supplement:

-Table S2: What is the land use history of these different protected areas? Have some areas been converted and are now recovering under protected status? Can you also comment in the text on differences in the ratio and potential efficacy of these protected areas under drought vs. non-drought years in reducing fire activity?

Unfortunately, a detailed land-use history for the different protected areas in Indonesia is not available. Previous work (Gaveau et al., 2013; Brun et al., 2015; Spracklen et al., 2015) shows protected areas consist of a diverse spectrum of primary and secondary forest with ongoing deforestation and degradation. We add the following text to the paper to comment on the difference between drought and non-drought years:

'Protected areas are less effective at reducing fire in drought years (e.g., 2015) compared to non-drought years.'

-Can you provide some further details on the designation/management practices between grand forest parks, national parks, nature reserves, etc.?

A detailed description of protected areas in Indonesia and their varied effectiveness is provided in Brun et al. (2015). We add the IUCN protected area categories to Table S2 and we add the following sentence to the Supplementary:

'A detailed description of the different protected area classifications is provided in Brun et al. (2015).'

Reviewer #2 (Remarks to the Author):

In this study the economic impact of peatland restoration is presented for Indonesian peatlands, following a number of possible scenarios where currently committed goals for peatland restoration had already been met.

This is a timely analysis as Indonesia's Peatland Restoration Agency is currently struggling to meet its restoration targets and appeared to be under resourced given the scale of the problem. The economic cost of failing to meet these targets are not necessarily obvious and it is great that these are being calculated and well presented as the authors have done. The hypothetical what-if scenario presented here is an elegant and scientifically valid way of demonstrating the costs that have been born through past inaction.

We thank the reviewer for their positive comments on our work.

While I support publication of this study, I would like to see better discussion of the assumptions contained within, none of which are necessarily unreasonable, but which I think could be better justified. I also have a number of smaller suggestions which I think could improve the manuscript.

We provided improved discussion of our assumptions as requested by the reviewer and described in detail in our point by point response below.

Main points:

- The authors argue for targeted restoration towards previously burnt sites, yet at no point was evidence provided to show that previously burnt sites are more likely to burn in subsequent years. The likelihood of repeat burning is something that needs to be established early on in the manuscript. Although I am aware this is often the case, I think it requires some support from the literature.

The following has been added to clarify why this method has been used:

'Fires are more likely to occur in areas made susceptible to fire by land use change and drainage^{54,55}, and so the locations of past fires are likely to be where future fires occur; areas of degraded peatland in Kalimantan have been found to have burned up to eight times between 1990 and 2011⁵⁶.'

- It is not yet fully clear what effect peatland restoration will have on fire prevention. I think in the short term, restoration is not likely to be so effective as to reduce fires in restored peatlands to the same degree as that of a natural system (as the authors have assumed), especially given that restored sites have been found to still have lower surface moisture contents compared to natural peatlands (ref 33, 34). However, given the current lack of data, I think it is acceptable to make this assumption, providing it is discussed and justified more than it is presently. I think it would suffice to briefly summarise what is and is not

known about the effect of restoration on fire suppression, both in terms of burn intervals and fire severity.

Thanks for this comment. We have expanded our discussion around peatland restoration and the use of protected areas as a proxy for restored areas:

‘Peatland restoration involves blocking drainage canals to restore water levels and re-establishing vegetation cover⁴⁷. Large-scale peatland restoration in Indonesia has just begun, and it is too early to measure the effect on fire⁴⁸. Instead, we used fires observed within protected areas as a proxy for fire occurrence on restored peatland. Peatland in protected areas is largely undrained and still covered in vegetation but is still subject to drought and anthropogenic pressures meaning that protected areas experience degradation, deforestation⁴⁹ and fire, albeit at a lower rate than surrounding unprotected land^{49–52}. Protected areas therefore provide a useful indication of the susceptibility of restored and re-wetted peatlands to fire under existing climate and anthropogenic pressures.’

- I don’t fully agree with the statement on line 223 where the authors state their analysis to be conservative in terms of burn scenarios, if restoration fails to reduce burn events to that of a natural system, which is entirely reasonable, then the analysis is not conservative. I suggest this is better justified or deleted

We agree that this statement may be confusing, as although the National Parks scenario is conservative compared to the other scenarios considered, it is possible that restoration may have a lesser impact on fires. We note that the National parks scenario does not assume a natural system, as National parks are still exposed to drought and anthropogenic pressures, meaning they still experience deforestation and fire. As suggested, we have removed this sentence and replaced with the following:

‘There is currently little data on the susceptibility of restored peatland to fire, and the effectiveness of peatland restoration interventions are limited to small-scale trials⁴⁸..’

Minor points:

- Given that fires are still occurring in protected peatlands and national parks, I can see why the no-burn scenario is presented as a less likely scenario. However, I think it is worth stressing that a no-burn scenario was the default state for the majority of Indonesia’s tropical peatlands prior to the last few hundred years. This is especially relevant if we are to use the present day burn rate in national parks as a ‘pristine’ reference point for restored sites. This paper may be useful <https://doi.org/10.1016/j.quascirev.2014.05.002>

We note that the present day fire occurrence in National parks is not meant to be the “pristine” reference state but a proxy for the reduction in fire that might be achieved through peatland restoration under anthropogenic pressures. The reviewer is correct that our “no-burn” scenario is the “pristine” reference state, but we consider that it is less likely given existing anthropogenic pressures and what is known about the difficulty in completely removing fire. We have revised our description of these scenarios to make this clearer.

We thank the reviewer for pointing us to this useful reference which we now cite in our revised manuscript.

- Figure 2: Do the authors mean 0.25 degrees latitude and longitude? (or both?). A legend for Kalimantan and Sumatra would improve this figure, as would annotating the plot to specify the blue percentage is reduction in burned area. At the moment there is rather a lot going on in this plot and the clarify would be improved by moving some information from the caption into the plot itself. The thick black line at zero would look better as a faint or dotted line, also the mean value for all protected areas in Kalimantan in 2015 looks to be exactly 1 which seems odd, the authors might want to check this data point.

The figure and figure caption have been updated as suggested, and the data checked. A legend for Sumatra and Kalimantan has been added and the line at $y=1$ is now dashed. We clarify that the blue percentage values are the reduction in burned area.

- Figure 5: Please define DALYS in the caption
This has been defined.

Reviewer #3 (Remarks to the Author):

Key results

The results presented in this paper are timely and highly relevant. The work demonstrates that taking action to prevent peatland forest degradation will have very positive impacts on both human health and carbon emissions resulting from anthropogenic activity. It provides a scientific basis to support land management policy in Indonesia. With a focus on the impact of wildfires and land clearing on peatland forests, this work also has relevance to other parts of the world, including north America and Canada.

We thank the referee for their positive comments on our work.

One weakness in the manuscript that I find concerning is the lack of emphasis put on loss of biodiversity and local subsistence livelihoods which are also significantly impacted by such fires. While it is understood that the analysis concentrates on 3 dimensions of fire impacts (land cover, carbon emissions and human health), I consider it vital that more statements should be included to ensure that the impacts on biodiversity and subsistence livelihoods are not neglected in the conclusions. Policy makers must not be led to believe that such impacts are trivial, even though it may be very difficult to assess them.

This is an important point and we agree that it is vital that policy makers consider these impacts and benefits. To address this important reviewer concern we have added a section to the discussion to discuss impacts on biodiversity, climate and subsistence livelihoods.

Validity

Notwithstanding the comments above, the data and analysis presented in this manuscript appears to present a valid statement about the geophysical impact of fires in Kalimantan and Sumatra. I am a little concerned however about the robustness of the financial and socioeconomic impacts which are based on less reliable data and tend to rely on more sweeping assumptions. For example, between lines 72 to 76, and other places in the

manuscript, and supplementary information, the 'costs' of fire damage are based on NPVs of each land use, sourced from a very limited number of publications which are some years old. It is not clear if the changes in price for the agricultural products mentioned over the period of years covered are taken into account. Also, a single value of forest seems to be used (supp info table S1). Furthermore, the conversion of these cited values to US\$ at the time of their publication perhaps do not represent the current value. In addition, these impact values are based on market prices and therefore do not include the consumer/producer surplus components of what makes up full economic value. To give more confidence in these financial estimates, the authors need to make it clear how these issues are dealt with.

Thanks for raising this concern. As suggested, we have revised our manuscript to make our assessment of the financial impacts more robust. We have synthesised NPV values for the different land uses from the literature and apply the average NPV for each land use, providing a much more robust assessment of the financial impacts. This means that for most land uses we are applying an average over multiple studies. For example, for palm oil we use an average value across 10 studies. For natural forest we apply a value based on 96 studies. We calculate financial impacts for the same year (2015) and adjust all values to 2015 US\$. We have updated the description of this in the Methods. More fundamental research is needed to improve the estimation of losses and damages from fires.

In figure 1 of the manuscript for example, the title suggests that these are factual values, while in reality they are estimates, and it should say that.

This has been added.

In that same figure, it differentiates health costs from Indonesia, from those in Singapore and Malaysia, and while that is important, it is not clear how those values from other countries are included in the total estimates for Indonesia (the focus of the paper). While this is discussed in the SI, this could be made clearer in the conclusions of the paper. (This could be an advantage from the study in that it provides insights into the problem for the neighbouring countries)

The paper is focused on Indonesian fires, and as such all economic losses associated with Indonesian fires have been included. As smoke is an international issue, fires in Indonesia can cause health impacts in neighbouring countries, which have then been included.

We add the following text to the conclusions as suggested:

“Peatland restoration in Indonesia has global benefits due to reduced CO2 emissions and regional benefits due to reduced air pollution.”

In relation to the health impacts, the use of Dalys from Non Communicable Diseases is of importance, although it must be noted that not all NCDs are the result of fires (eg diabetes and many forms of cancers). Perhaps this should be clarified in the text. Also, I would like to see reference to other publications rather than a total reliance on Keily et al for this part of the analysis. For example, the manuscript could benefit from citing other work such as:

- Proc Natl Acad Sci 115(38):9592–9597. <https://doi.org/10.1073/pnas.1803222115>
- Environmental Science and Pollution Research volume 26, pages 31315–31327 (2019)

- Sci Rep 6:37074. <https://doi.org/10.1038/srep37074>
- Shannon N Koplitz et al 2016 Environ. Res. Lett. 11 094023

We have added the following to reference the methods used and acknowledge the difference in mortality estimates for different studies:

‘Using relative risk functions at different PM_{2.5} concentrations is an established method for estimating the health impacts of PM_{2.5} exposure⁹⁹, and has been applied previously to Indonesian fire emissions^{26,71,100}. Using the GEMM and simulated PM_{2.5} concentrations, estimated premature mortality resulting from fires in 2015 is 56,000, lower than previous estimates (76,000 – 100,000)^{26,71} due to a less sensitive relative risk function²⁴. ‘

Furthermore, while Dalys are useful, they do only account for health burden rather than including a wider assessment of public health costs. No mention is made of the costs associated with short and long term impacts on mental health.

The reviewer is correct that we do not include impacts on mental health, as well established methods for these impacts do not exist. We add the following text:

“Our assessment of the health impacts of fire pollution excludes impacts on mental health and other wider health impacts⁷⁶”

Throughout the manuscript there seems to be some confusion between the terminology around costs, losses and values. This should be clarified. For example in lines 18/19, and elsewhere, the term ‘costs’ could be better replaced by ‘economic losses’. On line 80, the word ‘costs’ could be replaced by the term ‘imputed damage value of’. While some may think these points are trivial, they do have a significant meaning within the discipline of economics. Some further discussion on the section on costs of fires is provided below under methodology.

Thanks for these clarifications. We clarify our terminology as suggested.

Significance

It is my opinion that while I have outlined some weaknesses in the way the information is provided in the manuscript, I believe that this paper is an important addition to the relevant literature. It highlights the importance and benefits of large-scale landscape remediation and the means by which individual countries can contribute to reversing global degradation. I think it makes use of important scientific findings to support essential land use policy change, in particular the importance of developing and protecting national parks.

Thanks for these positive comments. We have highlighted and further emphasised the implications for land use policy, in particular the importance of developing and protecting national parks.

Data and methodology

The approach used in this manuscript takes advantage of some of the most recent advances in earth observation data to quantify the extent of fires in Indonesia. It combines this with an interesting method of evaluating the potential extent of fires in restored peatland areas. While the use of fire incidence and extent (and soil moisture levels) in protected areas may be a proxy for those that may occur in restored peatlands, the authors should clarify exactly how much of the protected areas included in the analysis are actually 100% peatland, as the

information in the text and on Table S2 only specifies the non-peatland areas of national parks, but does not explicitly state that the other types of protected areas are actually all peatland areas. (ie could be more explicit about this in section 2 of SI). When considering the effectiveness of Peatland restoration, the text (lines 173/174) suggests that data on soil moisture and burned area applied to all 51 hypothetical restoration areas is only drawn from Kalimantan, but these sites are distributed between Kalimantan and Sumatra, but conditions and impacts of restoration in the latter are different, as illustrated by Figure 2. Perhaps this should be clarified/justified if no other data from Sumatra is available.

We update the description to clarify that our assessment only includes peatland. We agree that there is uncertainty in the extent to which peatland restoration could reduce fire.

The reviewer is correct that we have used reductions in soil moisture and burned area from National Parks in Kalimantan as a proxy for soil moisture and burned area changes that could be achieved after peatland restoration across Sumatra and Kalimantan. The lower reduction in burned area in National Parks across Sumatra could be caused by differences in location, size, or management of the protected areas. We apply the burned area reductions observed in Kalimantan as an indication of the reduction in burned area that *can* be achieved through effective management. Our analysis highlights the needs for effective management of protected areas to prevent fire that is associated with logging, drainage and deforestation. We add the following sentence to the paper to clarify this point:
“Our analysis found varied effectiveness of protected areas at reducing fire, highlighting the need for effective management of protected areas⁸⁵ and peatland restoration.”

All protected area types are a mix of peatland and non-peatland, the burned area ratio for non-peat areas in national parks has been included in Table S2 as this is the only non-peatland value used in the study and referred to (in the methods, line 639). The caption for table S2 has been updated to reflect this.

While the use of the ‘no fire’ and ‘no peatlands fire’ scenarios are useful to investigate model uncertainty, it must be noted that there are many costly and practical difficulties in achieving the ‘no peatlands fire’ option in drought periods as large-scale rewetting (line 182) is not likely to be possible.

We agree that there are many challenges and barriers to achieving restoration. We have added a short discussion of this.

I am not an atmospheric scientist nor modeller and so am not fully qualified to evaluate the manuscript values on levels of CO₂ or particulate matter. I would however like to see more explicitly how these estimated impacts effect populations both near to or more distant from restoration sites (most of these likely being some distance from large human populations). Perhaps some kind of gradient figure could be provided since gridded population data is used.

Thanks for this suggestion. Our Figure 3 (e,f) (now c,d) provides an indication of the spatial extent of air pollution and the health impacts. Our estimate of the health impacts combines spatially explicit data on air pollution with spatially explicit data on population.

The maps shown in figure 3 are rather small and the similar colour palettes make it quite difficult to understand the information being presented therein. For example, is no account taken of potential DALYS occurring in Java as a result of these fires?

We have updated Figure 3 to only include 4 panels which we think are now clearer. We include the impact of potential DALYS across Indonesia including Java. As fire smoke tends to be carried West from Sumatra and Kalimantan during the dry season, there is little exposure to fire PM in Java.

In terms of the section on scaling up of restoration efforts, while this is an important section, it is highly unrealistic to ignore the important socioeconomic and institutional factors which would influence the success of such a policy (especially in Indonesia). For example, land ownership and cooperation from land users would be essential if such scaling up would be achieved. While this is briefly mentioned in the policy implications section (lines 306/307), I feel that it should be included more explicitly in this scaling up section, and perhaps repeated in the conclusions.

This is an important point. We have added a discussion of this in the section on scaling up as suggested. We add the following text:

'Rewetting and vegetation of peatlands are some of the most controversial solutions to peatland fire, with a major disconnect between resource users and policy makers⁸¹. Effective communication of the substantial health benefits delivered by peatland restoration could help build stronger support for restoration interventions from a wider spectrum of stakeholders⁸². Peatland restoration is likely to face serious socio-economic and cultural challenges that will constrain the scale of restoration that can be achieved.'

In the section on costs of fires, I have a number of concerns, both in terms of terminology/semantics and in terms of theoretical underpinning. For example, although the authors have discussed the issue of heterogeneity of fire damage, this has tended to be based on the geographical scale of fire areas rather than the heterogeneity within land cover types. Of particular concern is the lack of recognition of the imputed value of subsistence agriculture which often will be occurring within or on the margins of other commercial land uses, but is very rarely reported in any data or official reporting. The value of land cover damages as presented in table 1 of the SI and in the text (lines 372 to 381) are worryingly dependent on just two rather old references (refs 51 and 52) although a massive literature on this is available. I suggest some more references on land use values could be found. Again, the value of forest to local forest users and subsistence farmers is significantly underplayed in this manuscript (limited to pest control and pollination – mostly relevant to commercial farmers as these issues are mainly taken care of by the large number of plant types included in traditional tropical forest subsistence farms, eg in Amazonia, Vanuatu, Malaysia etc as well as in Indonesia). No attempt is made to include any costs associated with rewetting, although this is suggested as an option.

We have improved the terminology as suggested. We acknowledge that we are unable to correctly account for the value and contribution from subsistence agriculture. We have added a short statement to the discussion to acknowledge this. We have updated all the values for land cover damage to reflect more recent studies. Our values now represent an

average across a range of relevant studies reported in the literature. For example, our land value for palm oil is now an average across 7 studies. Our estimate of the cost of restoration includes the cost of rewetting.

In the section in lines 414 to 423, I find the very idea that there is any economic benefit from burning any forests, in particular tropical forests, is contrary to prevailing views. Since the loss of biodiversity, habitats and endangered species are amongst the highest in these forest areas, it is surely unwise to be suggesting that there is some level of economic benefit from burning them. Indeed the reference used (no 26) has been written from the perspective of fire as a tool for agriculture. It is wrong to use this to suggest that large-scale burning of forests is in any way a benefit to anyone or anything, except for the bottom line in unscrupulous large-scale forest and agribusinesses.

Why do the authors assume (eg on line 420) that there are any economic benefits from forest clearing, especially since they have failed to take account of biodiversity values, option values of potential pharmaceuticals etc. This point also questions the validity of the authors use of the term 'cost benefit analysis' (line 319 and ref 28), which cannot be correctly applied in this situation either practically (not enough available reliable values for all components of peatland forests) or theoretically (no clear mechanism by which all the considerations of welfare economics can be brought to bear on this problem. Maybe the term cost effectiveness analysis would be better as a true cost benefit analysis should highlight the processes by which the losers can be compensated by the winners from such calculations. While it is clearly of value to mention land buybacks (lines 277, 280), the lack of inclusion of biodiversity and subsistence values in this would render it inadequate or unworkable.

Thanks for these comments on our analysis. We agree that our terminology was not correct and we have changed this to reflect the reviewer comments. We know that fire is often used to clear land because it is cheaper than mechanical methods of land clearance. We now refer to the “reduced costs of using fire for land clearing” to describe this. We have removed all statements referring to the benefit of using fire.

Our reference to cost-benefit analysis on line 319 of the original manuscript refers to the cost-benefit analysis of peatland restoration (i.e., the costs of peatland restoration and the benefits of peatland restoration through a reduction in the losses and damages associated with fire). We have clarified this in the updated manuscript.

We agree that we have been unable to fully calculate the losses and damages caused by fire. We have added a discussion of the range of costs that we have not included, including biodiversity and subsistence livelihoods mentioned by the reviewer. Despite these missing costs, our analysis demonstrates that peatland restoration results in reduced losses and damages that exceed the costs of peatland restoration. That is the benefits of peatland restoration, through a reduction in the losses and damages of fire, exceed the costs of restoration. Our analysis, though incomplete, therefore provides strong support for peatland restoration.

Other points relating to the cost sections include:

- Has the changing value of the US\$ exchange rate been taken into account in different years (given land use values would be in Indonesian rupiah and then converted)?
We convert all data to US\$ (most studies already use US\$) and adjust data to 2015 values.

- It appears that there has been no attempt to include any estimate of the impact of fires on employment (+ or-)
We have added a statement to clarify that we do not attempt to include impacts on employment.

- Have the public health costs of treating health impacts of fires been included?
We focus on the main costs identified by the World Bank (Indonesia Sustainable Landscapes knowledge Note 1, World Bank), providing improved estimates across a range of years. We do not try and estimate impacts on employment or the public health costs of treating health impacts. We add a short discussion to the paper to acknowledge this.

Analytical approach

I believe I have covered this above.

Suggested improvements

- A very clear statement should be made at the start of the paper that not all costs and impacts of fires are included
Added as suggested.

- Could the title be improved? The paper does not include all the economic costs of Indonesian fires – perhaps it would be simpler and more accurate to have the title as: 'Assessing costs of Indonesian fires and benefits of restoring peatland'
Thanks for this suggestion. We have changed as suggested.

Clarity and context

- Clarification of terminology – eg costs vs losses (line 18/19 and elsewhere)
Revised and clarified as suggested.

- Need for greater transparency about what is not included in the analysis
We have added a discussion of what we do not include in this analysis.

- Would it not be better to put the policy implications and conclusions sections after the methods?
We follow journal formatting guidelines which place Methods at end of the paper.

References

In my opinion, more up to date references relating to costs of fires, values of forests and other land uses etc could be used.

We have updated references relating to the values of forests and other land uses and updated our calculation of costs.

Too much reliance on methodology of Keily et al in fire emissions inventory – ref to other approaches should be provided eg:

- Battye, W. and Battye, R (2002) Development of Emissions Inventory Methods for Wildland Fire Final Report February 2002. US EPA.
- van der Werf, G. R., Randerson, J. T., Giglio, L., van Leeuwen, T. T., Chen, Y., Rogers, B. M., Mu, M., van Marle, M. J. E., Morton, D. C., Collatz, G. J., Yokelson, R. J., and Kasibhatla, P. S.: Global fire emissions estimates during 1997–2016, *Earth Syst. Sci. Data*, 9, 697–720, <https://doi.org/10.5194/essd-9-697-2017>, 2017.
- Walker, X.J., Baltzer, J.L., Cumming, S.G. et al. Increasing wildfires threaten historic carbon sink of boreal forest soils. *Nature* 572, 520–523 (2019). <https://doi.org/10.1038/s41586-019-1474-y>

A reference to other fire emissions inventories has now been included:

'Fire emissions inventories often either exclude tropical peat fires⁷¹ or underestimate emissions¹⁴, due to difficulties in determining underground fuel consumed and tropical peat emission factors. Total FINNpeatSM PM_{-2.5} emissions in 2015 are a factor of 3.5 greater than those given by FINNv1.5, and 1.7 greater than those given by GFED⁷².'

My expertise

I am not able to fully evaluate the climatological and regional chemical modelling included here.

Professor Caroline A Sullivan, NSW Australia.

References

Hansson, A., Dargusch, P., An Estimate of the Financial Cost of Peatland Restoration in Indonesia, *Case Studies in the Environment* (2018) 2 (1): 1–8.
<https://doi.org/10.1525/cse.2017.000695>

REVIEWERS' COMMENTS

Reviewer #1 (Remarks to the Author):

The authors have addressed my comments in their revisions and I recommend for publication.

Reviewer #2 (Remarks to the Author):

I am fully satisfied with the response of the authors to my original comments, the assumptions of the study are now clear and well-justified. I am happy to recommend publication without any further changes.

I would also like to re-iterate that this is a very timely contribution to the issues around tropical peatland degradation. The authors have done an excellent job of communicating the costs associated with the status-quo and the failure to meet past restoration goals. I look forward to reading more of their research in the coming years.

Reviewer #3 (Remarks to the Author):

I have made a lengthy review of this paper and have examined the rebuttal provided by the authors and find their response thorough and appropriate. I am happy to support the publication of this paper in this revised format. I believe that the team has addressed all of the concerns that have been raised by all reviewers, and I look forward to seeing this important piece of work contributing to the literature through the nature platform

REVIEWERS' COMMENTS

We would like to thank all reviewers for their comments.

Reviewer #1 (Remarks to the Author):

The authors have addressed my comments in their revisions and I recommend for publication.

Reviewer #2 (Remarks to the Author):

I am fully satisfied with the response of the authors to my original comments, the assumptions of the study are now clear and well-justified. I am happy to recommend publication without any further changes.

I would also like to re-iterate that this is a very timely contribution to the issues around tropical peatland degradation. The authors have done an excellent job of communicating the costs associated with the status-quo and the failure to meet past restoration goals. I look forward to reading more of their research in the coming years.

Reviewer #3 (Remarks to the Author):

I have made a lengthy review of this paper and have examined the rebuttal provided by the authors and find their response thorough and appropriate. I am happy to support the publication of this paper in this revised format. I believe that the team has addressed all of the concerns that have been raised by all reviewers, and I look forward to seeing this important piece of work contributing to the literature through the nature platform